# Modeling of axonal endoplasmic reticulum network by spastic paraplegia proteins

Belgin Yalçın[1], Lu Zhao[1], Martin Stofanko[1], Niamh C O'Sullivan[1], Zi Han Kang[1], Annika Roost[1], Matthew R Thomas[1], Sophie Zaessinger[1], Olivier Blard[1], Alex L Patto[1], Anood Sohail[1], Valentina Baena[2], Mark Terasaki[2], Cahir J O'Kane[1]*

[1]Department of Genetics, University of Cambridge, Cambridge, United Kingdom; [2]Department of Cell Biology, University of Connecticut Health Center, Farmington, United States

**Abstract** Axons contain a smooth tubular endoplasmic reticulum (ER) network that is thought to be continuous with ER throughout the neuron; the mechanisms that form this axonal network are unknown. Mutations affecting reticulon or REEP proteins, with intramembrane hairpin domains that model ER membranes, cause an axon degenerative disease, hereditary spastic paraplegia (HSP). We show that *Drosophila* axons have a dynamic axonal ER network, which these proteins help to model. Loss of HSP hairpin proteins causes ER sheet expansion, partial loss of ER from distal motor axons, and occasional discontinuities in axonal ER. Ultrastructural analysis reveals an extensive ER network in axons, which shows larger and fewer tubules in larvae that lack reticulon and REEP proteins, consistent with loss of membrane curvature. Therefore HSP hairpin-containing proteins are required for shaping and continuity of axonal ER, thus suggesting roles for ER modeling in axon maintenance and function.

*For correspondence: c.okane@gen.cam.ac.uk

**Competing interests:** The authors declare that no competing interests exist.

## Introduction

Axons allow long-range bidirectional communication in neurons. They carry action potentials along their plasma membrane from dendrites and cell body to presynaptic terminals; and they transport cell components and signaling complexes using motor proteins, anterogradely and retrogradely. Another potential route for communication along axons is endoplasmic reticulum (ER); axons possess a network of ER tubules, which appears physically continuous both locally (*Tsukita and Ishikawa, 1976*; *Villegas et al., 2014*) and with ER throughout the neuron (*Lindsey and Ellisman, 1985*; *Terasaki et al., 1994*). The physical continuity of ER and its ensuing potential for long-distance communication has been likened to a 'neuron within a neuron' (*Berridge, 1998*).

Axonal ER appears mostly tubular and smooth, with some cisternae (*Tsukita and Ishikawa, 1976*; *Lindsey and Ellisman, 1985*; *Villegas et al., 2014*). Some rough ER is likely to be present too: numerous mRNAs are found in both growing and mature axons (*Zivraj et al., 2010*; *Shigeoka et al., 2016*), and local axonal translation can occur in response to injury (*Ben-Yaakov et al., 2012*; *Perry et al., 2012*). However, rough ER sheets (*Tsukita and Ishikawa, 1976*; *Villegas et al., 2014*), and markers of protein export and folding that are characteristic of rough ER (*Röper, 2007*; *O'Sullivan et al., 2012*), are relatively sparse in mature axons. Axonal ER therefore likely has major roles other than protein export; these could include lipid biosynthesis (*Tidhar and Futerman, 2013*; *Vance, 2015*), calcium homeostasis and signaling (*Ross, 2012*), and coordination of organelle physiology (*Phillips and Voeltz, 2016*). The continuity of the ER network suggests that some of these roles might have long-range as well as local functions; indeed, an ER-dependent propagating

**eLife digest** The way we move – from simple motions like reaching out to grab something, to playing the piano or dancing – is coordinated in our brain. These processes involve many regions and steps, in which nerve cells transport signals along projections known as axons. Axons rely on sophisticated 'engineering' to work properly over long distances and are vulnerable to diseases that disrupt their engineering. For example, in genetic diseases called 'hereditary spastic paraplegias', damages to the 'distal' end of axons – the end furthest from the nerve cell body – cause paralysis of the lower body.

Axons have several internal structures that make sure everything works properly. One of these structures is the endoplasmic reticulum, which is a network of tubular membranes that runs lengthwise along the axon. It is known that spastic paraplegias are sometimes caused by mutations affecting proteins that help to build and shape the endoplasmic reticulum, for example, the proteins of the reticulon and REEP families. However, until now it was not known how the ER forms its network in the axons and if this is influenced by these proteins.

To see whether reticulons and REEPs affect the shape of the endoplasmic reticulum, Yalçın et al. used healthy fruit fly larvae, and genetically modified ones that lacked the proteins. The results show that in healthy flies, the tubular network runs continuously along the axons. When either reticulon or REEP proteins were removed, the distal axons contained less endoplasmic reticulum. In mutant fly larvae that lacked both protein families, the endoplasmic reticulum was more interrupted and contained more gaps than in normal larvae. Using high-magnification electron microscopy confirmed these findings, and showed that the tubules of the endoplasmic reticulum in mutant axons were larger, but fewer.

A next step will be to test whether these mutations also affect how the axons work and communicate over long distances. A better knowledge of the role of the endoplasmic reticulum in axons will help us to understand how damages to it could affect hereditary spastic paraplegias and other degenerative conditions.

calcium wave is seen after axotomy of *Caenorhabditis elegans* or mammalian dorsal root ganglion neurons (*Ghosh-Roy et al., 2010*; *Cho et al., 2013*).

A strong hint of the importance of ER in axons is found in Hereditary Spastic Paraplegia (HSP), a group of axon degeneration disorders characterized by progressive spasticity and weakness of the lower limbs (*Blackstone et al., 2011*; *Blackstone, 2012*). Mutations affecting spastin, atlastin-1, reticulon-2, REEP1 and REEP2 account for most cases of autosomal dominant 'pure' HSP (*Hazan et al., 1999*; *Zhao et al., 2001*; *Züchner et al., 2006*; *Montenegro et al., 2012*; *Esteves et al., 2014*). These proteins share a common feature of one or two hydrophobic hairpin-loops inserted in the ER membrane, promoting ER membrane curvature in a process termed hydrophobic wedging (*Voeltz et al., 2006*). Proteins of the REEP and reticulon families localize preferentially to tubular or smooth ER, and their loss results in disruption of ER tubular organization (*Shibata et al., 2006*; *Voeltz et al., 2006*; *Park et al., 2010*; *Shibata et al., 2010*); they may also contribute to modeling of rough ER sheets by stabilizing their curved edges (*Shibata et al., 2009*).

What is the link between ER modeling and axon structure and function? HSP-causing mutations often appear to cause loss of protein expression or function (*Beetz et al., 2013*; *Novarino et al., 2014*), and the ability of hairpin-loop proteins to form homomeric and heteromeric complexes (*Shibata et al., 2008*) allows some point mutations to have dominant negative effects (*Züchner et al., 2006*; *Beetz et al., 2012*). Therefore loss of normal ER modeling appears to compromise axon maintenance and function. Given the roles of hairpin-loop proteins in ER modeling, we aimed to test the model that hairpin-loop-containing HSP proteins organize the axonal ER network. Since reticulon and REEP family proteins are redundantly required for most peripheral ER tubules in yeast (*Voeltz et al., 2006*), we focus on the requirement for these two families in axons. We previously showed that knockdown of the *Drosophila* reticulon Rtnl1 causes expansion of epidermal ER sheets, and partial loss of smooth ER marker from distal but not proximal motor axons (*O'Sullivan et al., 2012*). Here we show that REEP proteins have similar roles. We also show that

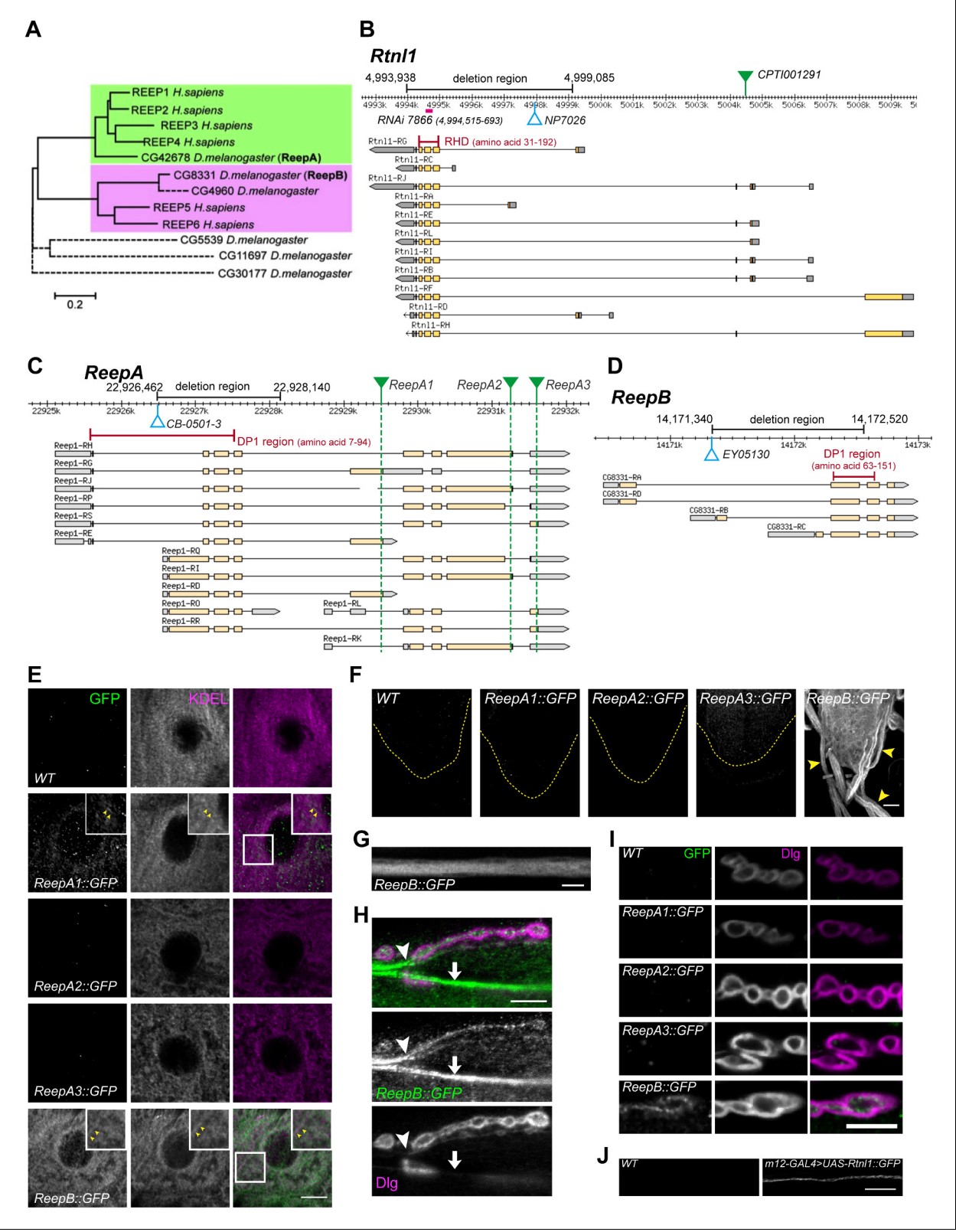

**Figure 1.** *Drosophila Rtnl1* and *REEP* genes and products. (**A**) A dendrogram based on ClustalW sequence alignment of *Drosophila* and human REEP proteins shows two branches corresponding to human REEP1-4 (CG42678) and human REEP5-6. Broken lines represent *Drosophila* REEP proteins that are evolving rapidly (***Supplementary file 1***), reflected by longer branch lengths. Sequences used are NP_075063.1, NP_057690.2, NP_001001330.1, NP_079508.2, NP_005660.4, NP_612402.1, NP_726266.1, NP_610936.2, NP_651429.1, NP_611831.2, NP_572730.1, NP_726366.1. (**B**) *Rtnl1* genomic and

**Figure 1 continued**

transcript map, showing the region deleted in *Rtnl1⁻* by excision of P-element *NP7026* (*Wakefield and Tear, 2006*; *Figure 1—figure supplement 1*), the RHD domain (Pfam 02453) and its coordinates in protein isoform G, the Rtnl1::YFP exon trap insertion *CPTI001291* (green triangle), and the fragment targeted by GD RNAi 7866. Map and coordinates are from the *Drosophila* Genome Browser (www.flybase.org, version R6.04), here and in subsequent panels; light regions in transcripts represent coding regions, dark shaded regions represent untranslated regions. (**C**) *ReepA* genomic and transcript map showing the region deleted in *ReepA⁻* by excision of P-element *CB-0501–3*, the position of the DP1 domain (Pfam 03134) and its coordinates in protein isoforms H and J. GFP insertion sites for *ReepA1::GFP, ReepA2::GFP* and *ReepA3::GFP* fusions are shown with green triangles. (**D**) *ReepB* genomic and transcript map showing the region deleted in *ReepB⁻* by excision of P-element *EY05130*, the position of the DP1 domain (Pfam 03134) and its coordinates in protein isoforms A and D). (**E–I**) Confocal sections showing localization of ReepA::GFP isoforms and ReepB::GFP. (**E**) Overlap of ReepA1::GFP and ReepB::GFP with anti-KDEL labeling in larval epidermal cells. To facilitate display of weaker ReepA1::GFP, the GFP channel in wild-type control (*WT*) and *ReepA::GFP* images has been brightened four times as much as for *ReepB::GFP*. (**F**) Expression of ReepA3::GFP and ReepB::GFP in third instar ventral nerve cord. The GFP channels for *WT* and *ReepA::GFP* have been brightened by twice as much as for *ReepB:: GFP*. VNCs are outlined with yellow dashed lines. Arrowheads show ReepB::GFP extending into peripheral nerves. (**G**) A single confocal section of a peripheral nerve, showing ReepB::GFP localized continuously along its length. (**H**) Double labeling of an NMJ for ReepB::GFP and the mainly postsynaptic marker Dlg, showing ReepB::GFP in an axon emerging from nerve bundles (arrowhead) to extend to the NMJ, as well as in axons traversing the muscle surface (arrow); note the dispersed ReepB::GFP staining also in the underlying muscle. (**I**) Double labeling of *ReepA::GFP* and *ReepB::GFP* lines for GFP and Dlg (mainly postsynaptic) shows presynaptic expression of ReepB::GFP. (**J**) GFP expression in a wildtype negative control (*WT*), or from Rtnl1::GFP expressed in two closely apposed motor neurons by *m12-GAL4*. Scale bars 10 µm, except F, 20 µm.

The following figure supplement is available for figure 1:

**Figure supplement 1.** Molecular lesions in *Rtnl1¹, ReepA⁵⁴¹* and *ReepB⁴⁸*.

---

simultaneous loss of reticulon and REEP family members leads to a range of axonal ER phenotypes, including a reduced network with fewer and larger tubules, and occasional gaps in the network. Our work implicates hairpin-loop-containing HSP proteins as important players in the axonal ER network, and suggests further models for how the network is organized.

# Results

## Two widely expressed REEP proteins, ReepA and ReepB, localize to the endoplasmic reticulum

The reticulon and REEP families of double-hairpin-containing proteins are collectively responsible for formation or maintenance of most peripheral ER tubules in yeast (*Voeltz et al., 2006*). We previously showed that the *Drosophila* reticulon ortholog Rtnl1 was strongly localized in axons, and that its knockdown caused partial loss of a smooth ER marker in posterior larval segmental axons (*O'Sullivan et al., 2012*). To test the roles of *Drosophila* REEP proteins in axonal ER localization, we first dissected the ortholog relationships between the six *Drosophila* and six human REEP proteins. Multiple sequence alignment of mammalian and *Drosophila* REEP protein sequences suggested that *CG42678* was the single *Drosophila* ortholog of mammalian *REEP1-REEP4* (*Figure 1A*). *CG42678* has previously been designated *Reep1* (http://flybase.org/reports/FBgn0261564.html), but we propose the name *ReepA* to reflect its orthology to the four mammalian genes *REEP1-REEP4*. Mammalian *REEP5* and *REEP6* appeared to share two *Drosophila* orthologs, *CG8331* and *CG4960* (*Figure 1*). We designated *CG8331* as *ReepB* because of its widespread expression (www.flyatlas. org; *Chintapalli et al., 2007*) and slower rate of evolutionary sequence divergence (*Supplementary file 1*). We excluded *CG4690* and three additional *Drosophila REEP* genes from further study, since their expression was restricted to testes and larval fat body (www.flyatlas.org; *Chintapalli et al., 2007*), and their faster rate of evolutionary sequence divergence (*Supplementary file 1*, and reflected in longer branch lengths in *Figure 1A*), suggesting poorly conserved function.

A Rtnl1::YFP exon trap, *Rtnl1^CPTI001291* (*Figure 1B*) was previously shown to localize to ER, including in axons (*O'Sullivan et al., 2012*). To study localization of ReepA and ReepB, we recombineered C-terminal GFP-tagged versions of these (*Figure 1C,D*) using P[acman] genomic clones (*Venken et al., 2009*). For ReepA, we generated EGFP fusions at three different C-termini (*Figure 1C*), that we called ReepA1::GFP (for protein isoforms D, E, G), ReepA2::GFP (for protein isoforms H, I, J, K) and ReepA3::GFP (for protein isoforms L, R, S). In epidermal cells, we detected

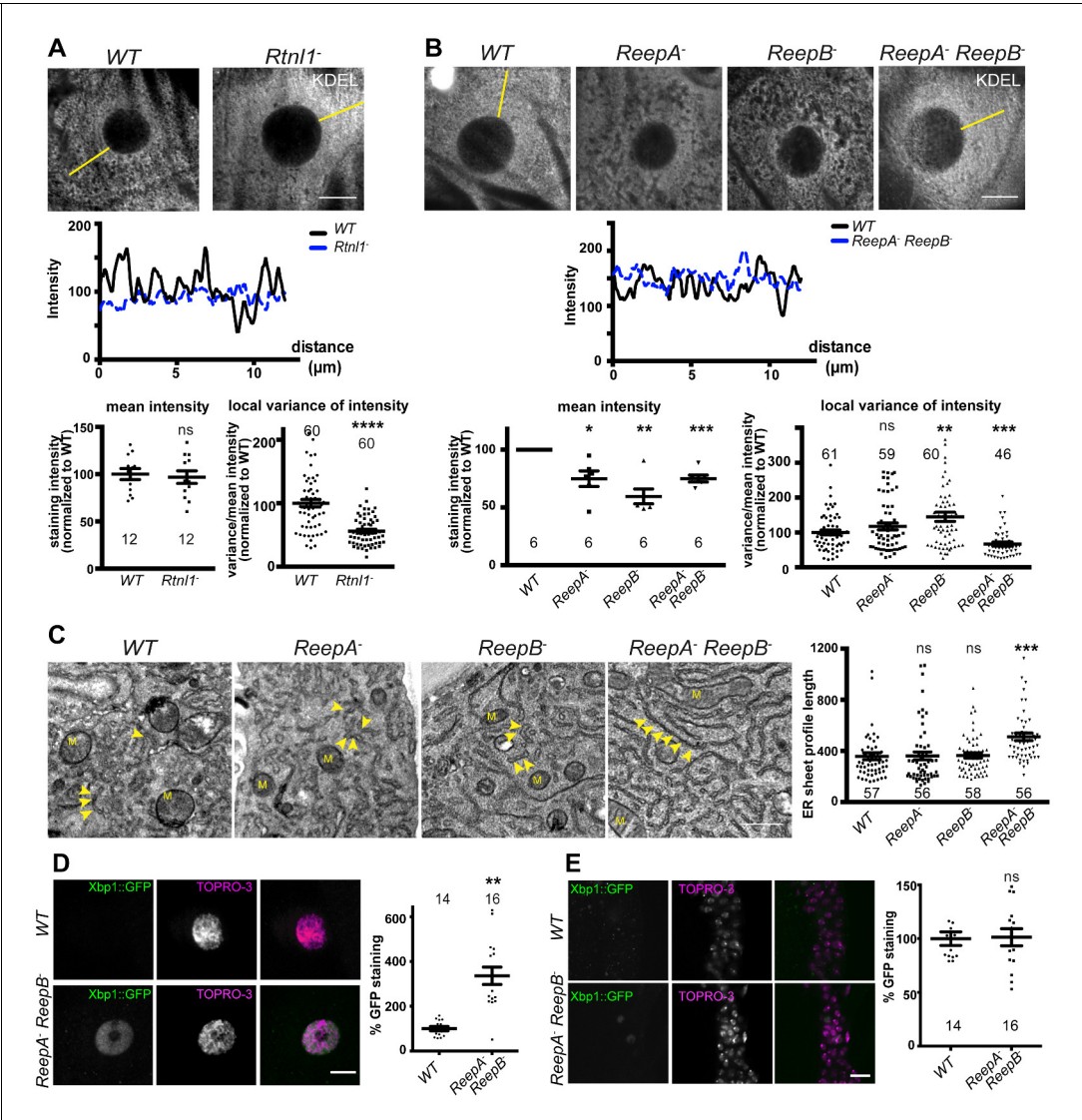

**Figure 2.** *Rtnl1⁻* mutants and *ReepA⁻ ReepB⁻* double mutants show disrupted ER organization in epidermal cells. (**A**) KDEL distribution in third instar larval epidermal cells appears diffuse in *Rtnl1⁻* larvae compared to a more reticular staining in wild-type (*WT*) larvae. Micrographs show 1.5 μm z-projections of confocal sections. KDEL intensity along yellow lines from nuclear envelope to the cell periphery shows less fluctuation in *Rtnl1⁻* (blue dotted line on line graph) than in *WT* larvae (black line on line graph). Fluctuation of intensity is quantified using the local normalized variance of intensity (variance/mean for rolling 10-pixel windows) over a 12 μm line for each cell. n = 61 epidermal cells from 12 larvae, 3–5 cells from each larva, from 3 independent experiments. (**B**) KDEL distribution in third instar larval epidermal cells shows less spatial fluctuation in *ReepA⁻ ReepB⁻* double mutant larvae compared to *ReepA⁺* (*WT*), and *ReepA⁻* and *ReepB⁻* single mutant larvae. This is confirmed by quantification as in **A**); overall KDEL intensity is reduced in all ReepA and ReepB mutant genotypes. n = 45–61 epidermal cells in total from 12 to 16 different larvae, 3–5 cells from each larva, from 6 independent experiments. Intensity datapoints are experimentwise averages of multiple larvae, compared by paired t-tests, since larva-wise data were too significantly different to pool across experiments in this case. (**C**) Electron micrographs show increased ER sheet profile length in *ReepA⁻ ReepB⁻* double mutant third instar larval epidermal cells compared to single *ReepA⁻* and *ReepB⁻* mutants and controls. Arrowheads, ER sheets; M, mitochondria. (n = 56–58 cells in total from three independent larvae, 18–19 cells from each. (**D,E**) *ReepA⁻ ReepB⁻* double mutant larvae have an increased ER stress response compared to a *ReepA⁺* (*WT*) control, measured by Xbp1::GFP expression, in larval epidermis (**D**) but not in neuronal cell bodies (**E**). Graphs show quantification of Xbp1::GFP staining intensity relative to controls. (n = 12–15 larvae). All graphs show individual datapoints and mean ±SEM. Occasional outlier datapoints off the top of the scale are omitted from graphs but included in statistical analyses. ns, p>0.05; *p<0.02; **p<0.005; ***p<0.0003; ****p<0.0001, two-tailed Student's t-test. Scale bars: A, B, D, E, 10 μm; C, 0.5 μm).

weak expression of ReepA1::GFP, but not of the other ReepA::GFP fusions; ReepB::GFP showed stronger expression still, and both fusions overlapped with an ER marker (KDEL; *Figure 1E*), similar to REEP proteins in other organisms (*Voeltz et al., 2006*; *Shibata et al., 2008*; *Park et al., 2010*). ReepB::GFP was also more strongly expressed in third instar larval CNS than ReepA3::GFP, which was the only ReepA::GFP fusion that we detected there (*Figure 1F*). ReepB::GFP, but no ReepA:: GFP fusion, was also detected in segmental nerves (*Figure 1F,G*), and in individual axons emerging from nerve bundles leading to the NMJ (*Figure 1H*). ReepB::GFP, but none of the ReepA::GFP fusions, localized as a mostly continual structure along the length of presynaptic terminals of neuro-muscular junctions (NMJs) (*Figure 1H,I*). We did not have a UAS-ReepB construct available, but *UAS-Rtnl1::GFP* (*Rao et al., 2016*) expressed in two adjacent motor neurons using *m12-GAL4* (*Xiong et al., 2010*) also showed strong axonal localization. We therefore conclude that ReepB::GFP localizes in axonal and presynaptic ER, similar to Rtnl1::YFP (*O'Sullivan et al., 2012*), but that none of the ReepA::GFP fusions is detectable in axonal or presynaptic ER.

An *Rtnl1* loss-of-function mutant, *Rtnl1*[1] (*Wakefield and Tear, 2006*), hereafter referred to as *Rtnl1*−, lacks the hydrophobic hairpin loop domain (RHD region) that induces ER membrane curvature (*Figure 1B*; *Figure 1—figure supplement 1*). We also used *P*-transposase-mediated imprecise excision to generate *ReepA*− and *ReepB*− mutants, both of which lack most of the curvature-mediating DP1 hairpin domains (*Figure 1B,C*; *Figure 1—figure supplement 1*).

## *Rtnl1*− and *ReepA*− *ReepB*− larval epidermal cells display an abnormal ER network

First, we asked whether *Drosophila* reticulon and REEP proteins contribute to ER network organization in third instar larval epidermal cells; *Rtnl1* knockdown leads to a more diffuse ER network organization and expansion of ER sheets in these cells, compared to wild-type controls (*O'Sullivan et al., 2012*). *Rtnl1*− mutant larvae also showed loss of ER network organization (*Figure 2A*). Intensity of KDEL labeling along a line from the nucleus to cell periphery displayed fluctuating intensity, reflecting the reticular distribution of KDEL in wild-type larvae, but less fluctuation in *Rtnl1*− larvae; overall levels of KDEL remained unchanged (*Figure 2A*). Similarly, loss of both *ReepA* and *ReepB*, but not loss of either gene alone, made KDEL levels in larval epidermal cells fluctuate less than in controls. Mean intensity of KDEL staining was decreased in all *ReepA* and *ReepB* mutant genotypes, and fluctuation in intensity was increased in *ReepB*− larvae (*Figure 2B*). Since confocal analysis suggests altered organization of ER in epidermal cells, but does not have sufficient resolution to reveal the details of how it is altered, we performed electron microscopy of epidermal cells. This revealed that *ReepA*− *ReepB*− double mutant epidermal cells showed longer ribosome studded sheet ER profiles, compared to controls, and to *ReepA*− and *ReepB*− single mutants (*Figure 2C*). This mutant phenotype is similar to, although less severe than loss of *Rtnl1* (*O'Sullivan et al., 2012*). *ReepA*− *ReepB*− mutants also showed increased ER stress in epidermal cells (*Figure 2D*) but not in CNS (*Figure 2E*). Therefore, Rtnl1 and REEP proteins shape the ER network in *Drosophila*, and their loss disrupts ER organization, causing longer ER sheet profiles.

## Loss of Rtnl1 or ReepB causes partial loss of axonal ER marker from posterior axons

To understand the roles of reticulon and REEPs in axonal ER organization, we labeled axonal ER by expressing Acsl::myc (*O'Sullivan et al., 2012*) in two adjacent motor neurons using *m12-GAL4* (*Xiong et al., 2010*). Loss of *Rtnl1* caused partial loss of Acsl::myc from posterior (segment A6) axons, but not from anterior axons (segment A2); it also caused Acsl::myc staining to appear more irregular in posterior but not anterior axons, reflected in a higher coefficient of variation (SD/mean) of Acsl::myc staining intensity along the length of posterior axons lacking Rtnl1, compared to wild-type (*Figure 3A*). These *Rtnl1* loss-of-function phenotypes were found either on targeted knockdown of *Rtnl1* in these motor axons or in *Rtnl1*− mutant larvae, and the *Rtnl1*− mutant phenotypes could be partially rescued by one copy of an *Rtnl1*[Pacman] genomic clone (*Figure 3A*).

*ReepA*− mutants showed no loss of Acsl::myc from axons. Similar to *Rtnl1* loss of function, *ReepB*− mutants showed partial loss of Acsl::myc from posterior but not anterior motor axons (*Figure 3B*); this phenotype was partially rescued with one copy of a genomic *ReepB::GFP* clone, and was also observed on *ReepB* knockdown in *m12-GAL4*-expressing neurons (*Figure 3B*). A

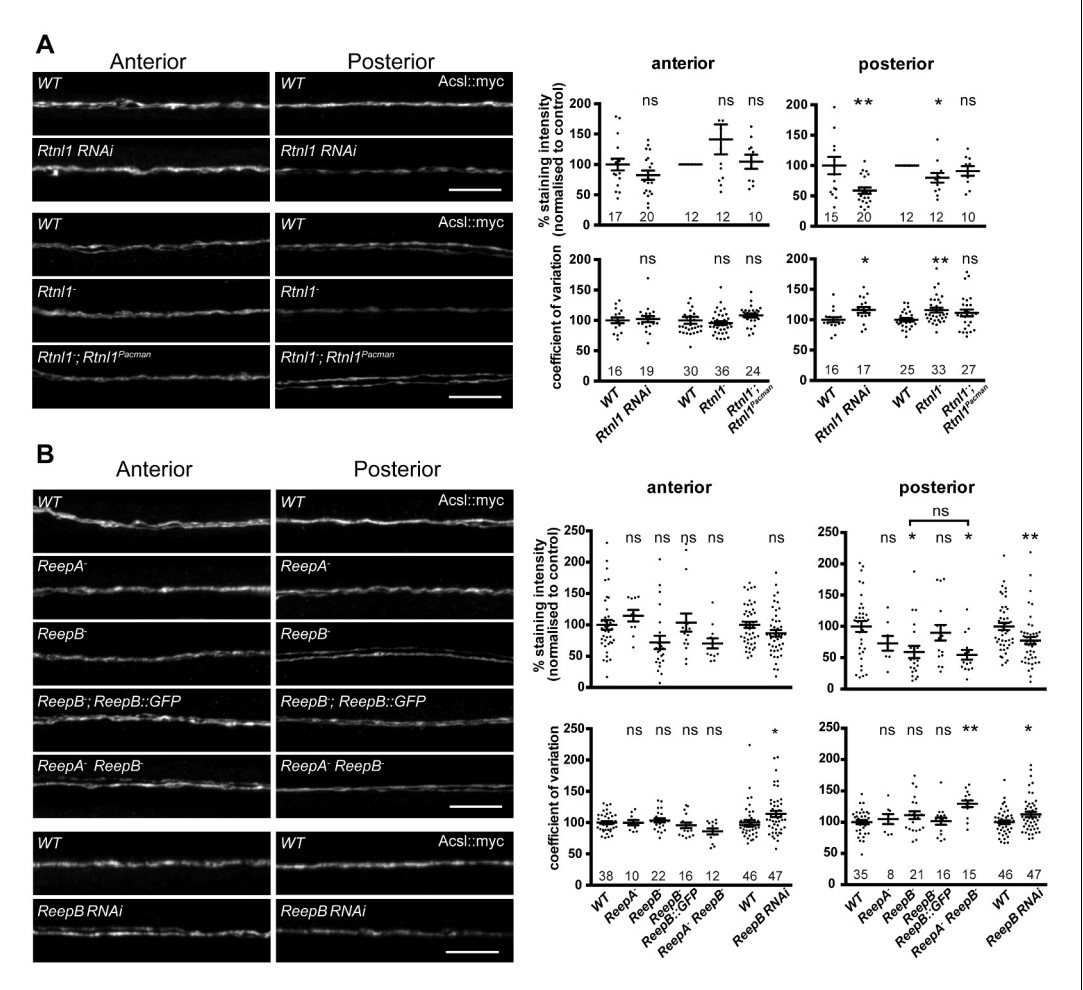

**Figure 3.** Loss of either Rtnl1 or ReepB leads to partial loss of smooth ER marker from distal motor axons. (**A**) Effects of Rtnl1 loss on ER, visualized by Acsl::myc expressed in two adjacent motor axons using *m12-GAL4*. Images show effects of *Rtnl1* knockdown, *Rtnl1⁻* mutation, rescue of *Rtnl1⁻* by *Rtnl1^Pacman* and respective *ReepA⁺* control (*WT*) axons. Graphs show mean staining intensity (top graphs) or coefficient of variation of intensity (bottom graphs) as a measurement of staining variability along the length of each axon. Rtnl1 loss leads to partial loss of Acsl::myc in posterior but not anterior axons (top graphs), and to some disorganization of posterior axonal ER seen by increased coefficient of variation of Acsl::myc staining intensity. There is partial rescue of *Rtnl1* phenotypes by one copy of a *Rtnl1^Pacman* genomic clone. n = 15–20 larvae per genotype pooled from 5 independent experiments for RNAi; n = 24–36 larvae per genotype from 10 to 12 independent experiments for *Rtnl1⁻* mutant. Graphs show individual datapoints with mean ±SEM; ns, p>0.05; *p<0.03; **p<0.005; two-tailed unpaired Student's t-test; two-tailed paired Student's t-test was used when larva-wise data were too significantly different across experiments (p<0.05, ANOVA) to pool. (**B**) Effects of ReepA or ReepB loss on axonal ER. *ReepB* knockdown or *ReepB⁻* mutation, but not *ReepA* mutation, causes partial loss of smooth ER marker Acsl::myc in posterior but not in anterior motor axons. The *ReepB⁻* phenotype can be mostly rescued by one copy of a genomic *ReepB::GFP* clone. A *ReepA⁻ ReepB⁻* double mutant shows a similar phenotype to a *ReepB⁻* single mutant. *ReepA⁻ ReepB⁻* double mutant, but not single *ReepA⁻* or *ReepB⁻* mutants, show increased coefficient of variation of Acsl::myc staining levels in posterior axons; n = 7–47 larvae from 4 to 11 independent experiments. All graphs show individual datapoints with mean ±SEM; occasional outlier datapoints off the top of the scale are omitted from graphs but included in statistical analyses. ns, p>0.05; *p<0.04 **p<0.01. *ReepB* RNAi was analyzed using two-tailed unpaired Student's t-tests; multiple comparisons of *ReepA* and *ReepB* mutant genotypes were analyzed by ANOVA followed by post-hoc Tukey HSD tests. Scale bars, 10 µm.

*ReepA⁻ ReepB⁻* double mutant showed loss of Acsl::myc from posterior axons, that was similar to that seen in *ReepB⁻* mutants (***Figure 3B***). *ReepA⁻ ReepB⁻* double mutant larvae, but not *ReepA⁻* single mutants, showed an increased coefficient of variation of Acsl::myc staining intensity in posterior but not in anterior axons; *ReepB* mutant and knockdown axons both showed a slight increase in coefficient of variation in posterior axons, although this was not significant for *ReepB* mutants (***Figure 3B***). In summary, loss of either Rtnl1 or at least ReepB alters axonal ER distribution in

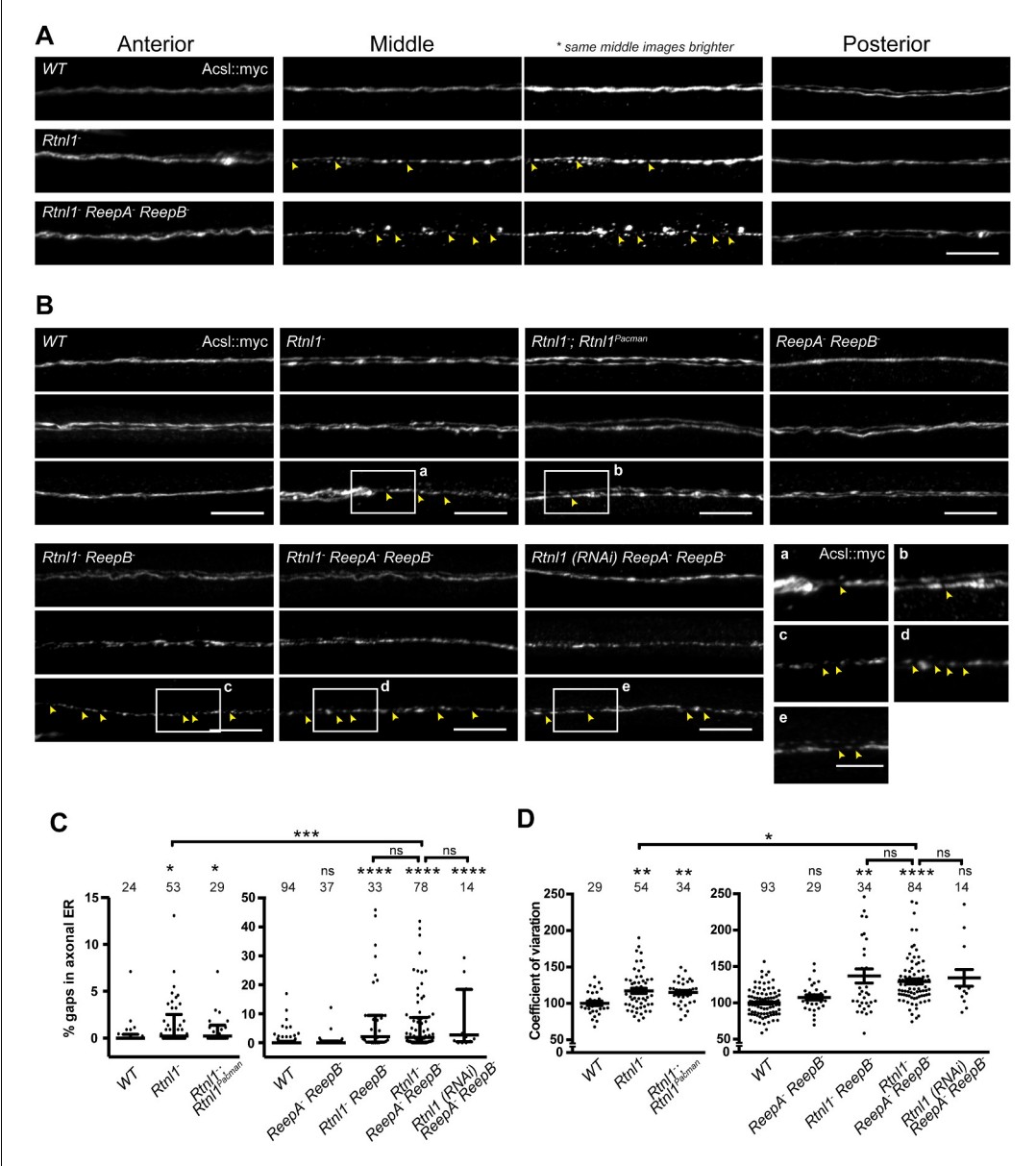

**Figure 4.** Loss of hairpin proteins leads to discontinuity of axonal ER staining. (**A**) *Rtnl1⁻* larvae and *Rtnl1⁻ ReepA⁻ ReepB⁻* triple mutant larvae sometimes show fragmented axonal ER labeling in the middle parts (segment A5) of long motor axons that express Acsl::myc under control of *m12-GAL4*. Anterior (segment A2) and posterior (segment A6) portions of the same motor axons show continuous ER labeling. Arrowheads show gaps in Acsl::myc staining; brighter versions of the same images show gaps in staining in mutants but not wild-type (*WT*), even when brightness of remaining staining is saturating. (**B**) Panels show three examples of the range of Acsl::myc distributions found in the middle parts of long motor axons of each genotype of *ReepA⁺* (*WT*), *Rtnl1⁻*, *Rtnl1⁻* rescued with a *Rtnl1^Pacman* construct, *ReepA⁻ ReepB⁻* and *Rtnl1⁻ ReepB⁻* double mutants, *Rtnl1⁻ ReepA⁻ ReepB⁻* triple mutant, and *Rtnl1(RNAi) ReepA⁻ ReepB⁻* larvae. A variety of phenotypes, from continuous to fragmented Acsl::myc labeling, are found in genotypes that lack Rtnl1, and tend to be more severe in genotypes that also lack ReepB. Insets show examples of gaps in ER continuity (arrowheads) at higher magnification. Scale bars 10 μm, and 5 μm in higher zoom images. (**C**) Percentages of a 45 μm length in the middle (A4/A5 segment) of each axon that lacks Acsl::myc staining, using an intensity threshold of 20 on a scale of 0–255. Individual axons are plotted, together with median and interquartile range; comparisons use Mann-Whitney U-tests. All genotypes that lack Rtnl1 show more gaps than *WT*; *Rtnl1⁻ ReepA⁻ ReepB⁻* triple mutants do not differ from *Rtnl1(RNAi) ReepA⁻ ReepB⁻* or *Rtnl1⁻ ReepB⁻* double mutants, but are significantly more severe than *Rtnl1⁻* single mutants. (**D**) The coefficient of variation of Acsl::myc labeling in middle axon portions is increased in genotypes lacking Rtnl1, relative to controls. Graphs show individual datapoints with mean ±SEM; occasional outlier datapoints off the top of the scale are omitted from graphs but included in statistical analyses. Comparison between *Rtnl1⁻* and *Rtnl1⁻ ReepA⁻ ReepB⁻* mutant genotypes was analyzed by two-tailed Student's t-test, other multiple comparisons by ANOVA followed by Dunnett's T3 test. ns, p>0.05; *p<0.04; **p<0.01; ***p<0.0005; ****p<0.0001; individual axons are plotted from 3 to 20 independent experiments, each with 2–3 different larvae for each genotype.

*Figure 4 continued on next page*

*Figure 4 continued*

The following figure supplement is available for figure 4:

**Figure supplement 1.** Plasma membrane integrity is not affected in *Rtnl1⁻ ReepA⁻ ReepB⁻* triple mutant larvae).

posterior motor axons, therefore supporting roles for these two protein families in axonal ER organization; in contrast loss of ReepA has either subtle or no effects.

## *Rtnl1⁻* and *Rtnl1⁻ ReepA⁻ ReepB⁻* mutants disrupt axon ER integrity and axon transport

Since loss of both reticulon and REEP families in yeast removes most peripheral ER tubules (*Voeltz et al., 2006*), we tested whether loss of both protein families in *Drosophila* might have similarly severe effects on axonal ER. *Rtnl1⁻ ReepA⁻ ReepB⁻* triple mutants were viable and fertile as adults, but survived poorly beyond two weeks of adulthood, compared to 4–5 weeks for wild-type adults. Triple mutant larvae showed increased fluctuation of Acsl::myc staining intensity along motor axons compared to wild-type, mainly in middle parts (segment A4 and A5) of longer axons. At its most extreme, this manifested as fragmentation of Acsl::myc labeling, which was never seen in wild-type axons (*Figure 4A,B*). When gaps in labeling were found in the central regions of axons, labeling in the anterior and posterior parts of the same axons was usually continuous (*Figure 4A*). Double labeling of plasma membrane (CD4::tdGFP) and axonal ER (Acsl::myc) showed no effect on axonal plasma membrane in *Rtnl1⁻ ReepA⁻ ReepB⁻* triple mutant compared to control axons (*Figure 4—figure supplement 1*), implying that the phenotype was limited to ER and did not affect axon integrity. To compare genotypes and labels, we quantified irregular labeling along a 45 µm stretch of axon traversing the larval A4/A5 region in two ways: first we measured gaps in labeling, defined as intensity below a threshold that could consistently distinguish labeled axons above background labeling in the nerve; and second, we quantified the coefficient of variation of labeling intensity along the axon (*Figure 4C,D*). *Rtnl1* loss-of-function axons, but not *ReepA⁻ ReepB⁻* mutant axons, also showed a mild ER fragmentation phenotype (*Figure 4A,B*). *Rtnl1(RNAi) ReepA⁻ ReepB⁻* larvae also showed a fragmentation phenotype similar to *Rtnl1⁻ ReepA⁻ ReepB⁻*, suggesting that loss of Rtnl1 is essential for it (*Figure 4B*). *Rtnl1⁻ ReepB⁻* double mutant axons also showed a similar phenotype to *Rtnl1⁻ ReepA⁻ ReepB⁻* triple mutants. Therefore, loss of Rtnl1 causes a mild irregular organization of axonal ER, and this phenotype is exacerbated by loss of ReepB, with no detectable contribution of ReepA loss; and loss of both Reep proteins has no apparent effect.

To confirm the physical continuity of ER in normal axons, and that gaps in Acsl::myc staining in *Rtnl1⁻ ReepA⁻ ReepB⁻* triple mutant axons reflect physical gaps in the ER network, we also visualized ER using fluorescent protein markers. The lipase CG9186::GFP (*Thiel et al., 2013*), Rtnl1::GFP (*Rao et al., 2016*) and tdTomato::Sec61β (*Summerville et al., 2016*) all localize to ER, and all showed continuous labeling in wild-type (*ReepA⁺*) axons (*Figure 5A*; *Figure 5—figure supplement 1A*). However, CG9186::GFP showed occasional gaps in triple mutant axons (*Figure 5A*), with at least one gap per larva visible by confocal microscopy in 6/17 larvae. Since confocal imaging might not reveal all physical discontinuities, and since gaps in marker distribution might not mean gaps in ER distribution, we probed the physical continuity of CG9186::GFP labeling using fluorescence recovery after photobleaching (FRAP). After bleaching a 12 µm length of axon in either wild-type axons or in *Rtnl1⁻ ReepA⁻ ReepB⁻* triple mutant axons lacking gaps, we observed rapid recovery of fluorescence from both ends of the bleached region (*Figure 5*; *Figure 5—figure supplement 1B*; *Video 1*), suggesting no physical barrier to CG9186::GFP diffusion. The kinetics of recovery were similar between wildtype and triple mutant axons (*Figure 5B–D*). We also performed FRAP on regions next to gaps in CG9186::GFP labeling in triple mutant axons (*Figure 5A*); here we observed good recovery of fluorescence from the end of the bleached region opposite the gap, but no recovery across the gap (*Figure 5*; *Figure 5—figure supplement 1C*; *Videos 2–4*), suggesting that gaps in CG9186::GFP labelling were also physical barriers to diffusion of CG9186::GFP, and hence gaps in the ER network.

During live imaging, we also followed the stability of the ER network in both wild-type and mutant axons. Features such as higher or lower levels of CG9186::GFP labelling (which could reflect

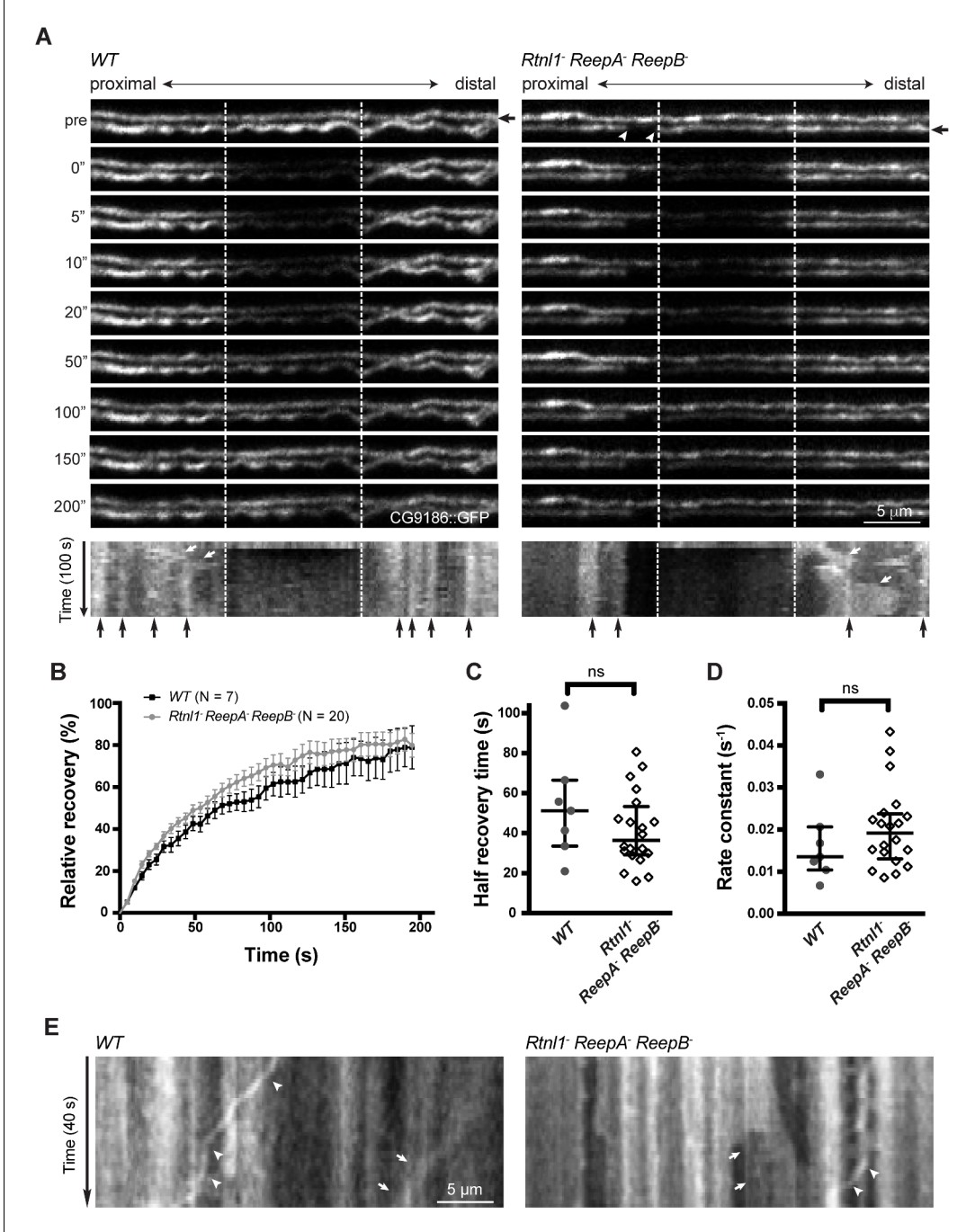

**Figure 5.** Live imaging of ER in wild type and hairpin mutant axons. (**A**) Representative FRAP assay images from *ReepA*[+] (*WT*) or *Rtnl1*[−] *ReepA*[−] *ReepB*[−] triple mutant axons in which ER was visualized using CG9186::GFP expression driven in two motor axons by *m12-GAL4*. One triple mutant axon shows a gap in labeling (arrowheads). Regions of interest (12 µm, between dashed lines) were photobleached. Fluorescence was visualized before photobleaching ('pre'), and during recovery over 200 s. A kymograph was generated for each axon indicated by an arrow in the top panel. Most areas of intense and less intense ER labeling remain stable over time (e.g. black arrows below kymographs); occasional movements of ER features are indicated by white arrows in kymographs. (**B**) Relative fluorescence intensities within the photobleached region were plotted (mean ±SEM) during recovery for wild-type axons or for triple mutant axons lacking ER gaps. (**C–D**) Quantification of half recovery time (**C**) and rate constant (**D**) for wild-type and triple mutant axons lacking gaps, with median and interquartile range. Data in **B–D** are from seven recordings from 4 wild-type larvae, and 20 recordings from 14 mutant larvae. ns, p>0.05; Mann-Whitney U test. Scale bars, 5 µm. (**E**) Representative kymographs from time-lapse recording of CG9186::GFP in unbleached single *ReepA*[+] (*WT*) or *Rtnl1*[−] *ReepA*[−] *ReepB*[−] triple mutant axons lacking gaps. The left panel shows retrograde movement of a ~ 3 µm length of ER labeling (arrows); the right panel shows anterograde movement of a ~ 5 µm stretch of ER labeling (arrows). Retrograde movements of labeled puncta are seen in both panels (arrowheads).

*Figure 5 continued on next page*

*Figure 5 continued*

The following figure supplement is available for figure 5:

**Figure supplement 1.** Continuity, stability and movement of fluorescent ER labeling in wild-type (*WT*) and *Rtnl1⁻ ReepA⁻ ReepB⁻* triple mutant axons.

variation in the numbers of tubules, or the presence of structures like cisternae) were mostly stable throughout a 200 s acquisition period (*Figure 5A,E*; *Figure 5—figure supplement 1*). However, we also observed some dynamic features, including anterograde or retrograde movement of more brightly labeled regions (*Figure 5A,E*) perhaps representing movement of ER tubules detached from the ER network, and retrograde movement of labeled puncta (*Figure 5E*; *Figure 5—figure supplement 1C*; *Video 3*), at around 0.3 μm/s. While most gaps in ER labeling in triple mutants were stable over 200 s (7 out of 8 gaps imaged, including *Videos 2–3*), in one case a gap was closed up by anterograde movement of a 5 μm length of ER proximal to the gap, which simultaneously opened up a new gap proximal to the moving section (*Figure 5—figure supplement 1C*; *Video 4*). In one case a labeled punctum moved retrogradely across an ER gap without pausing (*Figure 5—figure supplement 1C*), implying that microtubule-based transport was intact. Therefore, live imaging and photobleaching of CG9186::GFP both suggest that ER is normally continuous in axons, that loss of reticulon and REEP proteins leads to occasional gaps in ER labeling that represent physical breaks of ER continuity, and that ER network organization in axons shows both static and dynamic features.

We also tested for possible defects in axon transport by staining for abnormal accumulation of the synaptic vesicle protein CSP in axons. *Rtnl1⁻* larvae and *Rtnl1⁻ ReepA⁻ ReepB⁻* triple mutant larvae showed large accumulations of CSP in many peripheral nerves. The large accumulations of CSP in *Rtnl1⁻* larvae could be rescued by two copies of a *Rtnl1^Pacman* genomic clone (*Figure 6*).

## *Rtnl1⁻ ReepA⁻ ReepB⁻* mutant nerves show ER abnormalities in axons and glia

To better understand wild-type axonal ER organization, and the mutant phenotypes seen in confocal microscopy, we performed electron microscopy (EM) on 60-nm-thick serial sections of third instar peripheral nerves. Peripheral nerves contain both motor and sensory axons, arranged in fascicles, and wrapped in three main classes of glial cell (*Stork et al., 2008*; *Matzat et al., 2015*). We used ROTO staining (*Tapia et al., 2012*; *Terasaki et al., 2013*) to preferentially highlight cellular membranes including ER.

Wild-type larvae showed a network of ER tubules, in every axon that could be observed (*Figure 7A* left; serial sections in *Video 5*). Tubule outer diameter averaged around 40 nm (*Figure 7B,C*), and axons contained an average of around 1.6 ER tubules in each cross-section (*Figure 7D,E*). Reconstruction (*Figure 7F* left; *Supplementary file 2*) showed a tubular network with multiple branches, some dead ends, and continuity along nearly every axon sectioned. ER tubules often showed proximity to mitochondria or plasma membrane (*Figure 7G* left). We also found occa-

sional structures resembling small patches of ER sheets with an adjoining cisterna (*Figure 7H*), continuous with the tubular ER network (*Video 6*), at a frequency averaging around one per 20 μm per axon.

If *Rtnl1⁻ ReepA⁻ ReepB⁻* triple mutant larvae have less ER membrane curvature, we would expect them to have larger ER tubules, fewer tubules per section, loss of tubules, or a combination of these. EM revealed all these mutant phenotypes to varying degrees (*Figure 7*). ER tubule diameter was increased to around 60 nm (*Figure 7B,C*), allowing a lumen to be seen in some tubules (*Figure 7A* right; *Video 7*) that was rarely seen in wild-type (*Figure 7A* left; *Video 5*),

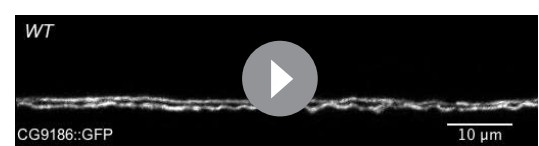

**Video 1.** Representative time-lapse microscopy of FRAP analysis in wild-type axons. Wild-type (*WT*) axons expressing CG9186::GFP under control of *m12-GAL4* were photobleached as described in *Figure 5A*. The video shows 2 s of pre-bleach images and 200 s of post-bleach images at one frame every 5 s. The bleached area is shown with a rectangle in the bleaching frame.

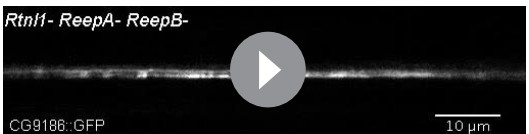

**Video 2.** Representative time-lapse microscopy of FRAP analysis in *Rtnl1⁻ ReepA⁻ ReepB⁻* triple mutant axons. *Rtnl1⁻ ReepA⁻ ReepB⁻* triple mutant axons expressing CG9186::GFP were photobleached and imaged as in *Video 1*, in a region immediately distal to a gap (between white arrows) in ER labeling in one axon.

**Video 3.** FRAP analysis in *Rtnl1⁻ ReepA⁻ ReepB⁻* triple mutant axons showing a CG9186::GFP labeled particle passing retrogradely across the gap region. *Rtnl1⁻ ReepA⁻ ReepB⁻* triple mutant axons expressing CG9186::GFP were photobleached and imaged immediately distal to an ER gap as in *Video 2*. A labeled particle (white arrow) can be seen moving retrogradely across the ER gap (white arrowhead).

and most triple mutant axons exhibited only a single ER tubule (*Figure 7D,E*). Reconstructions (*Figure 7F* right; *Supplementary file 3*) showed a less extensive ER network in mutant axons. Frequent contacts of ER with mitochondria and plasma membrane were found in both wild-type and *Rtnl1⁻ ReepA⁻ ReepB⁻* mutant axons (*Figure 7G*). Both wild-type and mutant axons showed swellings containing mitochondria and clusters of vesicles resembling synaptic vesicles (*Figure 7I*; *Video 8*). We did not detect large swellings with synaptic vesicle materials similar to those seen with confocal microscopy (*Figure 6*), but this may reflect the much shorter lengths of nerve examined by EM, around 5 µm compared to around 50 µm in confocal.

Serial EM sections also revealed variable fragmentation of ER in *Rtnl1⁻ ReepA⁻ ReepB⁻* mutant axons (*Figure 7J,K*), consistent with that seen using confocal microscopy (*Figure 4*). Some discontinuity of the ER network was observed in about 10% of wild-type or *Rtnl1⁻ ReepA⁻ ReepB⁻* mutant axons (*Figure 7L*). However, gaps in *Rtnl1⁻ ReepA⁻ ReepB⁻* mutant axons (*Figure 7J,K*; *Video 9*; *Supplementary file 4*) were longer (*Figure 7M*), and slightly more numerous (but not significantly when averaged across larvae; *Figure 7N*) than in wild-type axons, resulting in nearly a four-fold increase in the length of affected axons that lacked ER tubules (*Figure 7O*).

*Rtnl1⁻ ReepA⁻ ReepB⁻* mutant peripheral nerves also showed glial cell phenotypes. Wild-type peripheral nerves are surrounded by an outer perineurial glial cell, and just beneath this a subperineurial glial cell; axons or axon fascicles are wrapped imperfectly by a wrapping glia cell (*Stork et al., 2008*; *Matzat et al., 2015*). All glial classes, but particularly subperineurial glia, showed a trend towards increased ER sheet profile length compared to control cells (*Figure 8A–I*), similar to *Rtnl1* knockdown (*O'Sullivan et al., 2012*) or *ReepA⁻ ReepB⁻* double mutant (*Figure 2*) epidermal cells. Triple mutant wrapping glia also displayed more extensive wrapping, sometimes completely ensheathing axons, which was rarely observed in control nerve sections (*Figure 8J–M*; *Figure 8—figure supplement 1*).

## Discussion

The existence of a tubular axonal ER network has been known for decades. Nevertheless, the cellular mechanisms that organize a compartment that is usually distributed throughout cells, along the great lengths of axons, are until now largely unknown. The finding that several causative genes for the axon degenerative disease HSP encode ER modeling proteins, suggests a link between ER modeling and axon function or maintenance, and provides candidate proteins that may be instrumental in structure and function of the axon ER network. These candidates include

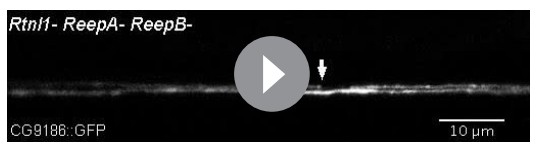

**Video 4.** FRAP analysis in *Rtnl1⁻ ReepA⁻ ReepB⁻* triple mutant axons showing dynamic ER gap generation. *Rtnl1⁻ ReepA⁻ ReepB⁻* triple mutant axons expressing CG9186::GFP were photobleached and imaged immediately distal to an ER gap as in *Video 2*. Anterograde movement of a 5 µm length of ER simultaneously closes up a gap distal to it (first white arrow) and opens up a new gap (second white arrow) proximal to it. A kymograph from this preparation is shown in *Figure 5—figure supplement 1C*.

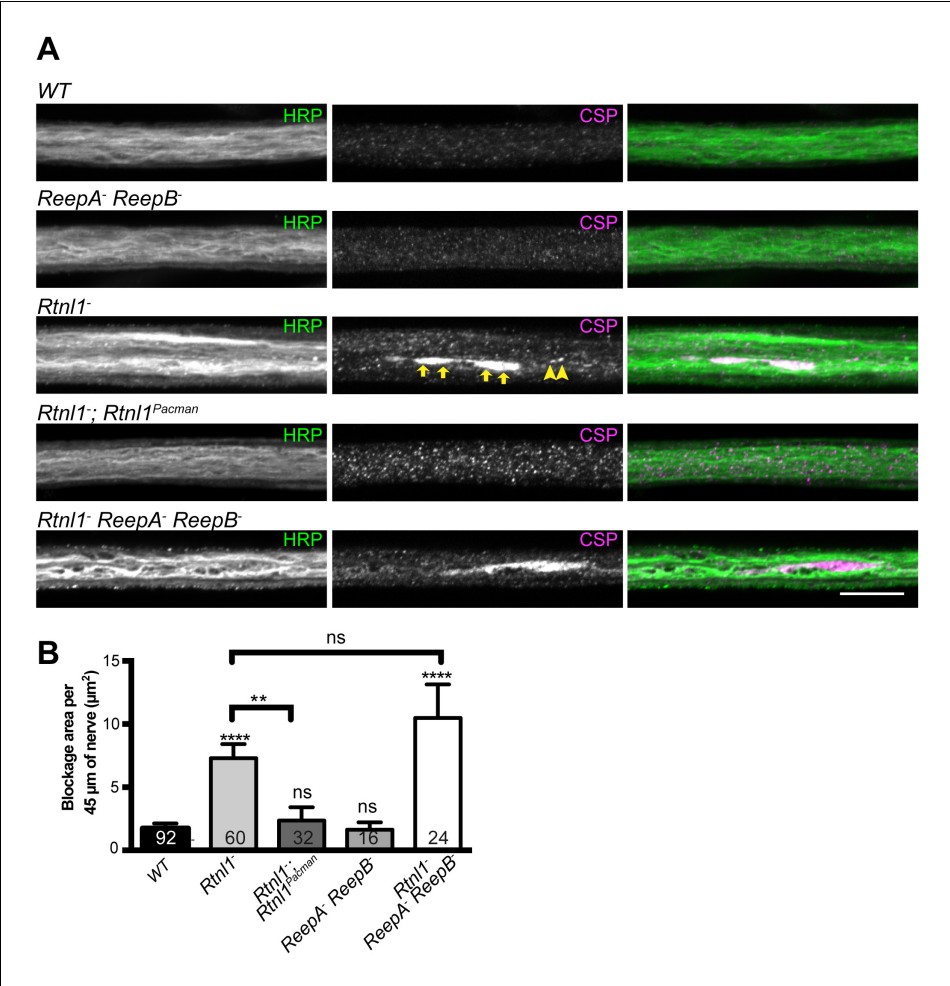

**Figure 6.** Loss of Rtnl1 causes mild accumulation of synaptic vesicles in axons. (**A**) Peripheral nerves of *Rtnl1⁻* and *Rtnl1⁻ ReepA⁻ ReepB⁻* triple mutant larvae show larger accumulations of synaptic vesicle protein CSP (e.g. yellow arrows), and smaller elongated CSP puncta (e.g. yellow arrowheads). In contrast, control larvae (*WT*) and *ReepA⁻ ReepB⁻* double mutant larvae show an even distribution of small round CSP puncta. CSP accumulations are not significantly bigger in *Rtnl1⁻ ReepA⁻ ReepB⁻* triple mutants than in *Rtnl1⁻* mutants. The CSP accumulations in *Rtnl1⁻* larvae can be rescued by two copies of a *Rtnl1^Pacman* genomic clone. All axons shown are crossing abdominal segment A2. (**B**) Graph shows mean ±SEM; n = 16–92 axons from 8 to 46 larvae, from 3 different experiments. ns, p>0.05; **p<0.006; ****p<0.0001, two-tailed Student's t-test. Scale bar, 10 μm).

several hairpin-loop-containing HSP proteins, of the spastin, atlastin, reticulon, REEP, and Arl6IP1 families, that influence ER structure in situations including yeast, mammalian cultured cells, and neuronal cell bodies in vivo (*Shibata et al., 2006*; *Voeltz et al., 2006*; *Hu et al., 2008*; *Shibata et al., 2009*; *Park et al., 2010*; *Shibata et al., 2010*). The HSP-related protein families that model ER, and some other proteins that interact with them, share a common feature of one or two intramembrane hairpin loops that can insert into the cytosolic face of the ER membrane, thereby recognizing or inducing curvature. This property makes the reticulon and REEP(DP1) families together responsible for most peripheral ER tubules in yeast, and contribute to the curved edges of ER sheets (*Voeltz et al., 2006*; *Hu et al., 2008*). The latter property may explain the expansion of ER sheets in *Drosophila* lacking the reticulon Rtnl1 (*O'Sullivan et al., 2012*) and in *REEP1* homozygous mutant mice (*Beetz et al., 2013*).

Given this background, we set out to test how far the reticulon and REEP families contribute to axonal ER organization. *REEP1* homozygous mutant mice were not previously tested for effects on axonal ER, although knockdown of *Drosophila Rtnl1* led to partial loss of smooth ER marker in distal

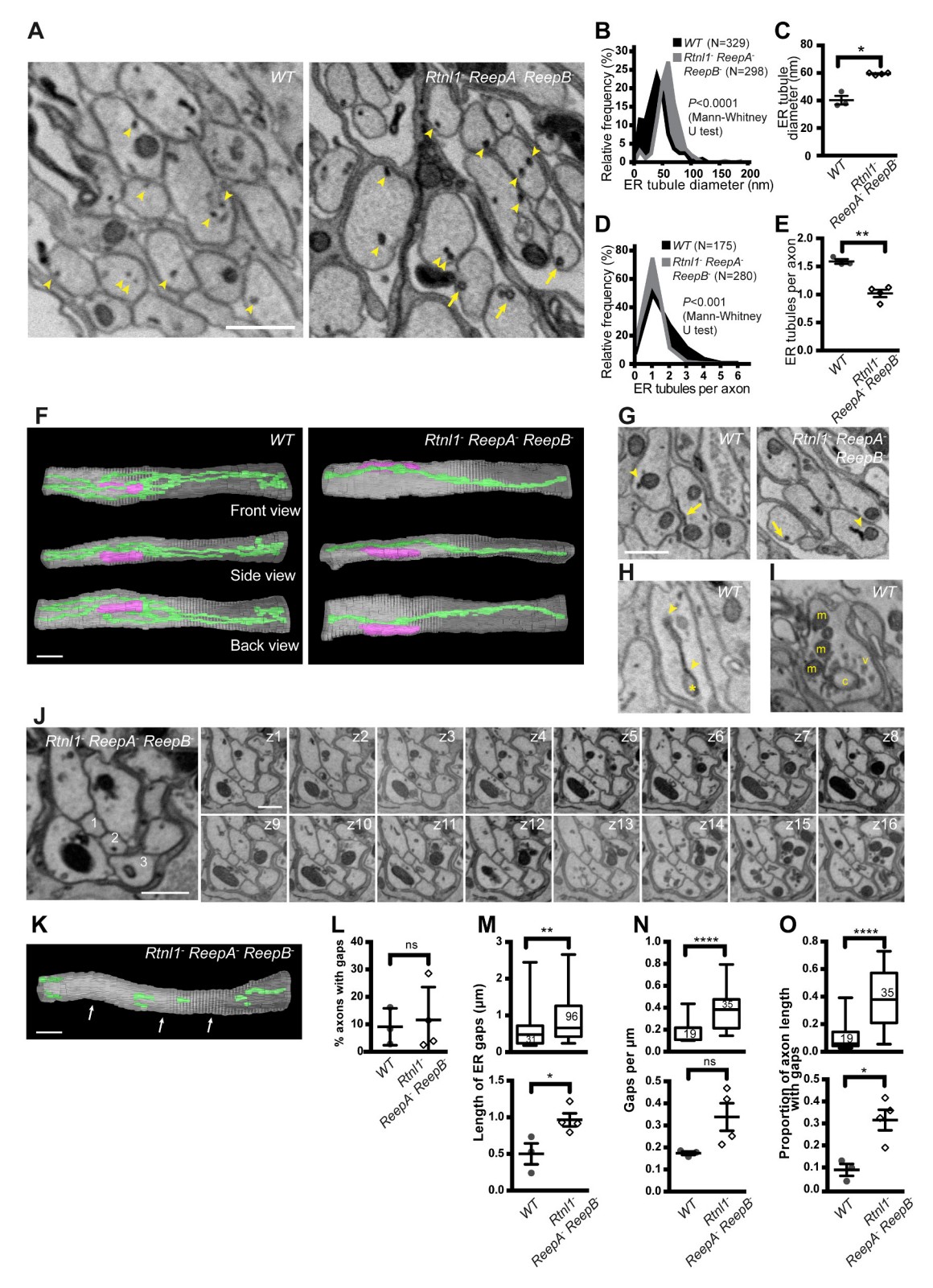

**Figure 7.** Loss of hairpin proteins leads to fewer but enlarged ER tubules in larval peripheral nerve axons. (**A**) EMs of peripheral nerve axons from wild-type *ReepA⁺* (*WT*, left) or *Rtnl1⁻ ReepA⁻ ReepB⁻* triple mutant (right) larvae. Arrowheads indicate ER tubules (seen as continuous structures in serial sections in *Videos 5* and *7*). Triple mutant axons show enlarged ER tubules, sometimes with a clear lumen (arrows), seldom observed in wild-type. Quantification of ER tubule diameter (**B,C**) and tubules per axon cross-section (**D,E**) in control and mutant larvae. Data from individual ER tubules or

*Figure 7 continued on next page*

*Figure 7 continued*

axons are shown in B and D, averaged larval values, mean ±SEM in C and E. (**F**) 3D reconstruction of a 4.5 µm axon segment from wild-type (left) or mutant (right) peripheral nerves, generated from 75 serial 60 nm sections, showing ER (green), mitochondria (magenta) and plasma membrane (gray). Interactive versions of the reconstructions are in *Supplementary files 2* and *3*. (**G**) Electron micrographs of peripheral nerve axons from wild-type (*ReepA⁺*, left) or *Rtnl1⁻ ReepA⁻ ReepB⁻* triple mutant (right) larvae, showing proximity of ER to mitochondria (arrowheads) or plasma membrane (arrows). (**H**) Representative EM of short ER sheet (arrowhead) and cisterna (asterisk) from a wild-type larva; further sections in *Video 6*. (**I**) section of an axonal swelling from a wild-type larva showing mitochondria (m), vesicles (v), and a large clear cisterna (c); further sections in *Video 8*. (**J**) Serial EM sections show ER discontinuity in two mutant axons: axon 1 lacks ER tubules in sections z6-z14 and axon 3 in sections z1-z11; neighboring axons (e.g. axon 2) show a continuous ER network. (**K**) 3D reconstruction of a 4.5 µm axon segment from mutant peripheral nerves, generated from 75 serial 60 nm sections, showing multiple gaps (indicated by arrows). ER is in green and plasma membrane in gray. Raw EM data and an interactive version of the reconstruction are in *Video 9* and *Supplementary file 4*, respectively. (**L**) Frequency of axons with gaps in the 4.5 µm lengths analyzed. ER gap length (**M**), numbers of ER gaps per µm (**N**), and proportion of axon length with gaps (**O**) in affected axons. In M–O, top graphs show data from individual ER gaps (**M**) or individual axons (**N–O**), with second and third quartiles and 5th and 95th percentiles; bottom graphs show averaged larval values, mean ±SEM. In all graphs, ns p>0.05; *p<0.04; **p<0.003; ****p<0.0001; Mann-Whitney U test for **B**, **D**, top graphs in **M–O**; two-tailed Student's T test for **C**, **E**, **L**, bottom graphs in **M–O**. Scale bars, 500 nm.

motor axons (*O'Sullivan et al., 2012*). Here we build on this work by analyzing mutants of all the widely expressed and highest conserved members of the reticulon and REEP families in *Drosophila*: Rtnl1, an ortholog of all four human reticulons (*O'Sullivan et al., 2012*); ReepA, an ortholog of human REEP1-REEP4; and ReepB, an ortholog of human REEP5-REEP6. We monitored phenotypes of these mutants, singly and in combination, by confocal microscopy of axonal ER markers in small numbers of motor neurons, live imaging and photobleaching of ER, and EM using membrane-specific staining.

Confocal microscopy revealed a partial loss of ER marker in distal but not in more proximal motor axons in *Rtnl1* and in *ReepB* mutants. Although the only *REEP* genes identified as causative for HSP are *REEP1* and *REEP2*, loss of their ortholog *ReepA* had at most only mild effects on axonal ER (*Figures 3* and *4*). The stronger axonal ER phenotypes of *ReepB* or *Rtnl1* loss of function, compared to *ReepA⁻* mutants, is consistent with the higher levels of *ReepB* and *Rtnl1* expression, judged by transcriptomics (www.flyatlas.com), and the detection of ReepB and Rtnl1 but not ReepA fusions in peripheral nerves (*Figure 1F,G*; *O'Sullivan et al., 2012*) or individual axons (*Figure 1H,J*). The par-

tial loss of distal axonal ER marker in both mutants and motor-neuron knockdowns, of either *Rtnl1* or *ReepB* (*Figure 3*; *O'Sullivan et al., 2012*), and qualitatively similar ER fragmentation phenotypes in *Rtnl1 ReepA ReepB* triple loss-of-function genotypes, obtained using either *Rtnl1* mutants or motor-neuron knockdown (*Figure 4*), suggest that the axonal ER phenotypes are cell autonomous in motor neurons.

Given the joint and partly redundant requirement of the reticulon and REEP families for ER tubule formation in yeast (*Voeltz et al., 2006*), we tested whether this was also true for axonal ER. Flies lacking Rtnl1, ReepA and ReepB – probably equivalent to mammals lacking all four reticulons and all six REEPs, and homozygous viable – indeed showed more extreme ER phenotypes than axons lacking either Rtnl1 or ReepA and ReepB alone (*Figures 3* and *4*). However, only a fraction of mutant axons showed severe fragmentation, and even affected axons still had continuous labeling of axons with ER marker through much of their length. ER fragmentation in the middle parts of long axons might be a consequence of axon expansion during larval growth,

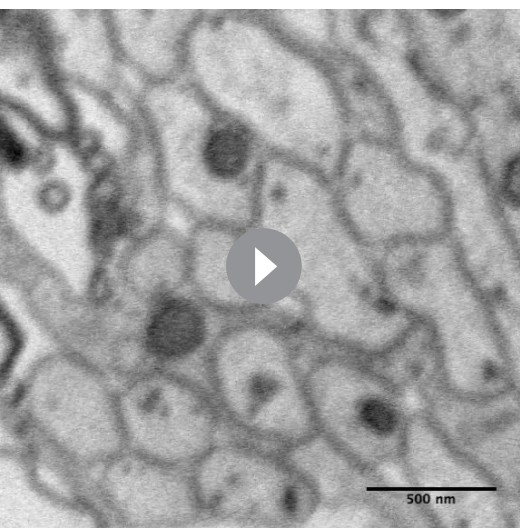

**Video 5.** Serial EM sections of a wild-type peripheral nerve, showing continuity of tubular membrane structures through multiple sections. The sixth section in the series is shown in *Figure 7A* (left). ER tubules were identified as darkly stained structures present for multiple sections. See *Figure 7A* for annotations.

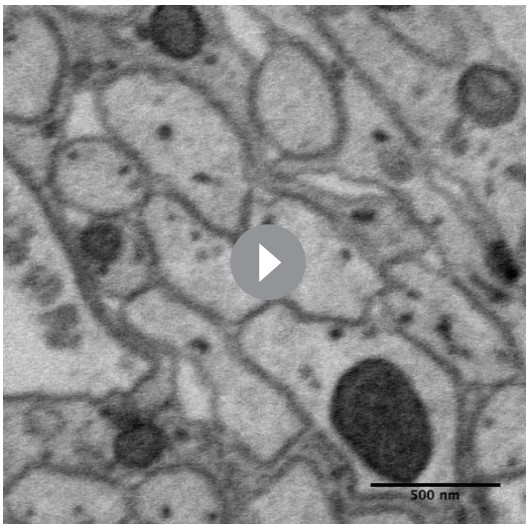

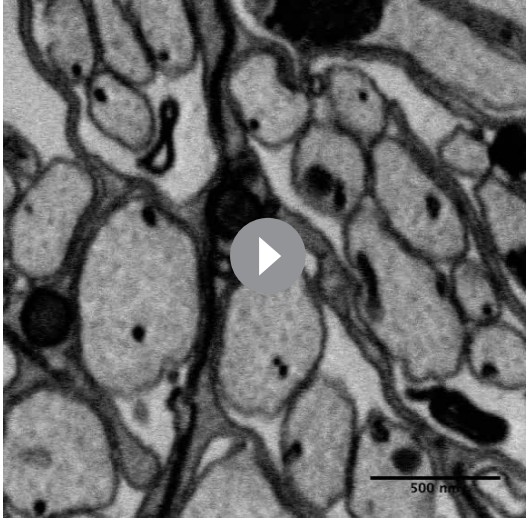

**Video 6.** Serial EM sections of a sheet-like ER structure in a wild-type axon, shown in *Figure 7H*. Arrows indicate short ER sheets and associated cisternae, present across multiple sections. Note that these structures usually appear continuous with tubules in adjacent sections.

**Video 7.** Serial EM sections of a *Rtnl1⁻ ReepA⁻ ReepB⁻* peripheral nerve showing continuity of tubular membrane structures through multiple sections. The sixth section in the series is shown in *Figure 7A* (right). ER tubules were identified as darkly stained structures present for multiple sections. See *Figure 7A* for annotations.

in which the somatic and presynaptic ends of the axon are gradually pulled apart, with insufficient ER-modeling proteins to maintain the expanding tubular network throughout the axoplasm. Therefore reticulon and REEP proteins are present in axons and have roles in ER organization there – but since triple mutant axons still mostly possess ER, there must be additional proteins required too.

These might be found among the increasing number of other HSP genes that encode ER proteins with possible hairpins, such as *Arl6IP1/ SPG61*, which affects ER organization (Novarino et al., 2014; *Yamamoto et al., 2014*; *Fowler and O'Sullivan, 2016*), or *C19orf12/SPG43* (*Landouré et al., 2013*). The variable nature of the triple mutant fragmentation phenotype might reflect stochastic variation in the amounts of such proteins, the amount of ER present, external factors like physical stresses during larval movement, or dynamic fluctuations in the local levels and connections of ER tubules. The tubular ER network is highly dynamic in non-neuronal cells (*Nixon-Abell et al., 2016*; *Valm et al., 2017*) and in axons our live imaging suggests dynamic features superimposed on a structure that is largely stable over the 1–2 min of imaging (*Figure 5*).

EM examination of wild-type axonal ultrastructure revealed that ER tubules were effectively ubiquitous in *Drosophila* peripheral nerve sections, as seen previously in mammalian neurons (*Tsukita and Ishikawa, 1976*; *Villegas et al., 2014*), albeit with fewer tubules, presumably reflecting the smaller diameters of the axons

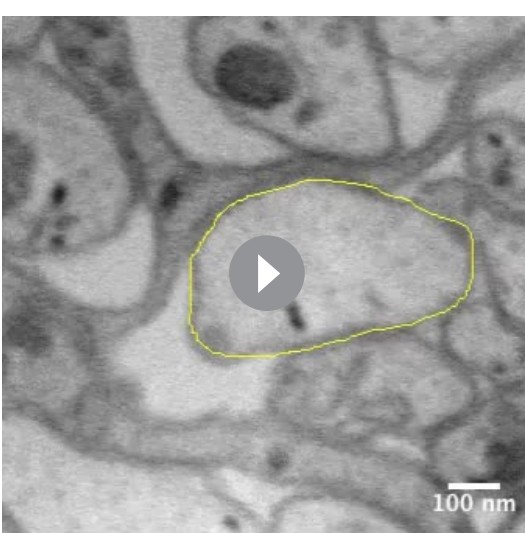

**Video 8.** Serial EM sections of a wild-type axonal swelling shown in *Figure 7I*. An axon with a swelling is highlighted in the first frame. Sections show accumulated mitochondria, vesicles, and a large clear cisterna, as labeled in *Figure 7I*.

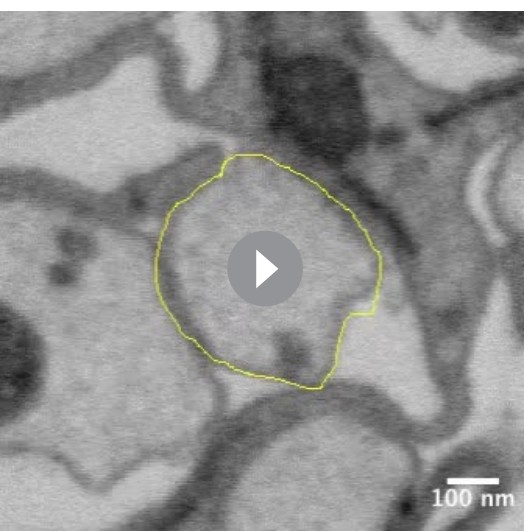

**Video 9.** Serial EM sections of a 4.5 μm segment of a *Rtnl1⁻ ReepA⁻ ReepB⁻* mutant axon with disrupted continuity of ER, used for 3D reconstruction in **Figure 7K**. An axon with gaps in its ER network is highlighted in the first frame. Continuous ER tubules were identified as the presence of signals at the same position for three or more sections. Given the varying brightness and contrast of EM sections, faint staining that coincided with a tubule signal in adjacent sections was also considered as an ER tubule. Complete loss of ER tubules from three or more sections was defined as a gap.

examined here. Reconstruction over several μm showed a continuous network of ER tubules in most axons examined, in agreement with the ER continuity found in neurons by lipid dye labeling (*Terasaki et al., 1994*), and recently in reconstructions of serial sections from focused ion beam SEM (*Wu et al., 2017*). However, in a few axons, we found short lengths of axon with no detectable ER (*Figure 7J–O*). There could be several reasons for this: *Terasaki et al. (1994)* only assessed continuity in dendrites, cell body and proximal axon; some of the gaps we observe in EM could be short transient gaps in a dynamic network; some of the apparent gaps could be 'thin ER' observed using higher-resolution focused-ion-beam SEM (*Wu et al., 2017*), but that might be missed using our approach; larval axons with low diameters might be intrinsically more susceptible to occasional gaps in the ER network, than wider axons with more tubules; and we might occasionally miss an ER tubule due to weaker staining, or close proximity to other structures like plasma membrane. We also observed occasional small ER sheet-like structures in wild-type axons (*Figure 7H*; *Video 6*). Although we do not see ribosomes on these, this could be due to lack of staining by the ROTO protocol. As discussed above, rough ER and translation are relatively sparse in axons, but low levels of rough ER are possible, and consistent with the occasional sheet structures observed here.

EM also showed phenotypes consistent with loss of ER membrane curvature in *Rtnl1⁻ ReepA⁻ ReepB⁻* triple mutant axons (*Figure 7*). Mutant axons had ER tubules of larger diameter, fewer tubules per axon cross-section, and consistent with our confocal data (*Figure 4*), longer gaps in the ER network than wild-type, although most parts of most mutant axons examined still had a continuous ER network (*Figure 7*). These mutant phenotypes could potentially have physiological consequences. Larger tubules could potentially store and release more calcium than thinner ones, while the reduced network could make the role of the ER in calcium buffering or release more localized. The less extensive ER network in mutants might also reduce the amount of contact between ER and other organelles, with consequences for calcium and lipid homeostasis that require these contacts, or for regulation of mitochondrial fission (*Friedman et al., 2011*) – although the continuing proximity of ER to mitochondria and plasma membrane in mutants means that any effects are presumably quantitative rather than qualitative. The reduced curvature of ER membrane in mutants might also influence their protein composition, since many membrane proteins have mechanisms for recognizing differential membrane curvature (*Antonny, 2011*). The occasional lack of continuity could prevent propagation of ER-dependent $Ca^{2+}$ signals like those seen in injured mammalian sensory neurons (Cho et al., 2013); it could also cause local impairments in $Ca^{2+}$ or lipid homeostasis that could lead to local transport inhibition, as is the case for mitochondrial transport (*Wang and Schwarz, 2009*), although lack of ER continuity does not appear to directly prevent axon transport (*Figure 5—figure supplement 1C*; *Video 3*). Sporadic lack of ER continuity might explain the preferential sensitivity of distal longer axons to HSPs, since these would be more likely to suffer from a gap in ER continuity to the cell body, compared to proximal or shorter axons. In this model, disease-causing alleles in single hairpin-encoding genes could promote degeneration in distal motor axons by increasing the probability of such gaps, dependent on factors such as age, axon length or diameter, and ER tubule density and dynamics.

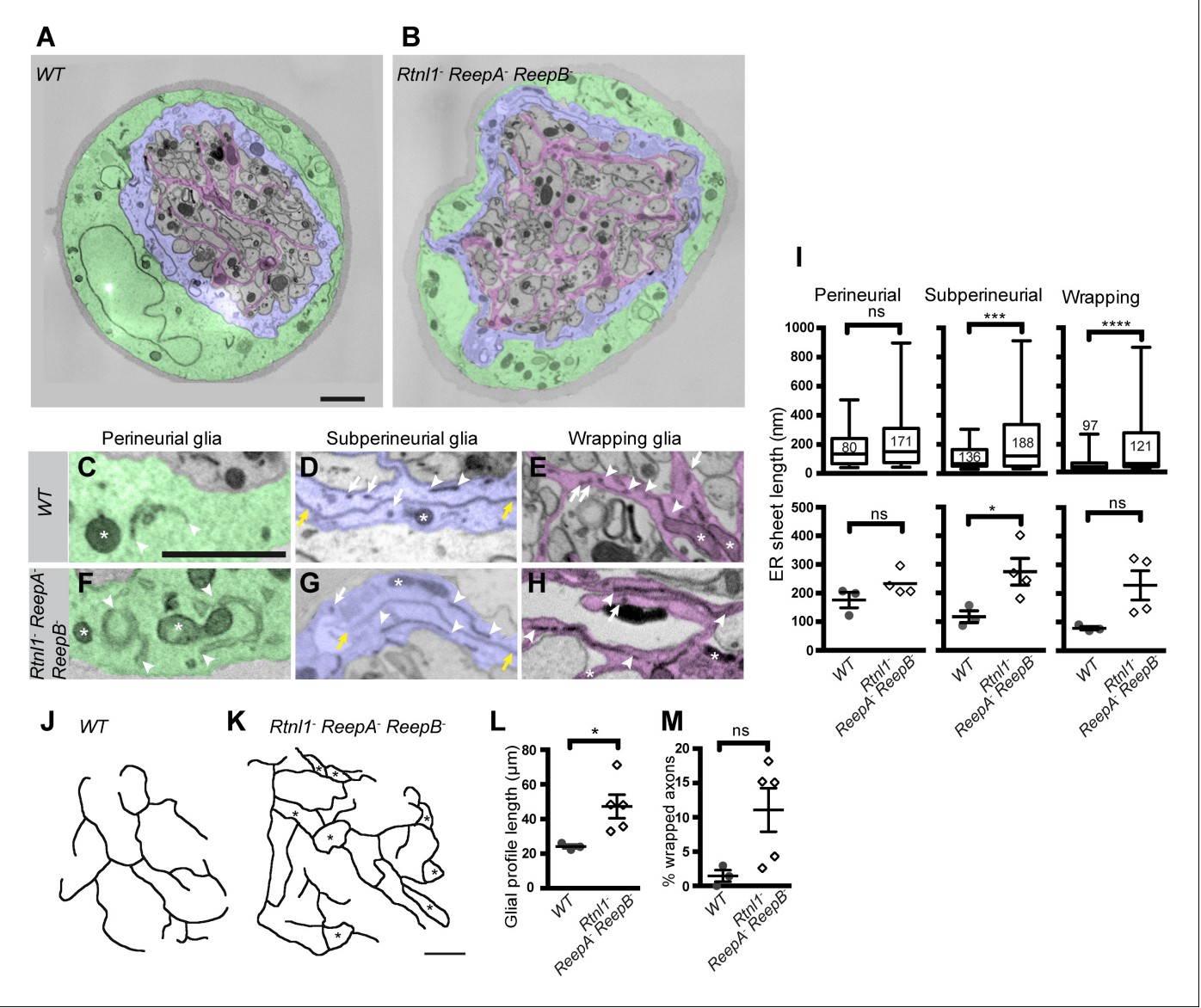

**Figure 8.** Loss of reticulon and REEP proteins leads to ER disorganization in glial cells and hyper-wrapping of peripheral axons. (A–B) EMs of peripheral nerve sections from *ReepA+* (*WT*) (A) or *Rtnl1− ReepA− ReepB−* triple mutant (B) larvae. Perineurial, subperineurial, and wrapping glial cells are shaded green, blue, and magenta, respectively. (C–H) Higher magnification images of perineurial (C,F), subperineurial (D,G), and wrapping (E,H) glia from wild-type (C–E) or mutant (F–H) nerve sections, showing ER tubules (white arrows; confirmed as tubules by presence in adjacent sections) and sheets (white arrowheads). Note the longer ER sheet profiles and fewer ER tubules in the subperineurial (G) and wrapping glial cells (H) of mutant nerves. Asterisks show mitochondria; yellow arrowheads show glial plasma membrane, identified by its continuity. (I) ER sheet profile length in control and mutant glial cells from 3 wild-type and 4 mutant larvae. Top graphs represent all individual sheet profiles, with second and third quartiles and 5th and 95th percentiles; bottom graphs show averaged larval values, showing mean ±SEM. Two-way ANOVA showed a significant effect of genotype (p<0.002) but not glial class (p>0.3) on ER sheet length, with no interaction between factors (p>0.3). (J,K) Sketches of wild-type (J) and mutant (K) wrapping glial cells, showing excess processes in the triple mutant. Asterisks indicate completely wrapped one-axon or two-axon fascicles, rarely seen in wild-type nerve sections. More examples of each phenotype are in *Figure 8—figure supplement 1*. (L–M) Quantification of wrapping glial membrane profile length per nerve cross-section (L) and percentage of axons that are wrapped individually or as two-axon fascicles (M) in wild-type and mutant nerve sections (mean ±SEM). ns, p>0.05; *p<0.05; ***p<0.001; ****p<0.0001. Mann-Whitney U test (I), top graphs); two-tailed Student's t-test (I), bottom graphs; L, M,). Scale bars, 1 μm.

The following figure supplement is available for figure 8:

**Figure supplement 1.** Sketches of wrapping glia in wild-type (*ReepA+*) and *Rtnl1− ReepA− ReepB−* mutant peripheral nerves, similar to those in *Figure 8J,K*.

The apparent ubiquity of ER in axons, the extent of its continuity over long distances, and the preferential susceptibility of distal longer axons to mutations that affect ER-modeling proteins, all point to important physiological roles of this compartment and of its continuity. In this work we have begun to reveal the mechanisms that determine its organization. We have shown roles for two protein families that contain HSP disease gene products, in influencing the shape of individual tubules and the axonal ER network, with potential physiological consequences that would also be affected by mutations in these genes. Understanding the consequences of axonal ER structural defects for ER dynamics and axonal physiology, both in the genotypes we have described here, and in other genotypes that might also affect axonal ER organization, will provide models for the potential physiological defects in HSP and other axon degeneration diseases.

## Materials and methods

### *Drosophila* genetics

$ReepA^{541}$ (referred to as $ReepA^-$), and $ReepB^{48}$ (referred as $ReepB^-$) mutants were generated by imprecise excision of *P* elements *CB-0501–3* (RRID:DGGR_123207) and *EY05130* (RRID:BDSC_16636) shown in *Figure 1*. A precise excision generated in these experiments, $ReepA^{+C591}$ (referred to as $ReepA^+$) was used as a genetic background control where feasible. $Rtnl1^1$, referred as $Rtnl1^-$, was a gift from G. Tear (*Wakefield and Tear, 2006*; FlyBase ID FBal0246222). $Rtnl1^-$, $ReepA^-$ and $ReepB^-$ recombinants were generated by meiotic recombination on the second chromosome, and recombinants were screened using PCR primers (*Supplementary file 5*) to diagnose wild-type or mutant alleles of all three genes. Mutant and wild-type stocks were frequently genotyped to ensure that experimental flies were not contaminated. For knockdown experiments, either *UAS-Rtnl1-RNAi* line 7866 (construct GD900, which has no predicted off-targets; FlyBase ID FBti0098310), or the $w^{1118}$ control stock, 60000 (both obtained from the Vienna *Drosophila* RNAi Center, www.vdrc.at), or the *UAS-ReepB-RNAi* line 8331 R-3 (National Institute of Genetics Fly Stock Center, Japan; FlyBase ID FBal0275953) was crossed with *UAS-Dcr2; CyO/If; m12-GAL4, UAS-Acsl::myc. UAS-Dcr2* (RRID:BDSC_24648; *Dietzl et al., 2007*) was also present for knockdown in $Rtnl1$ (RNAi) $ReepA^-$ $ReepB^-$ larvae. Other fly stocks used were *P{UAS-Acsl.715.MycC}3* (RRID:BDSC_32330; *Zhang et al., 2009*), *PBac{681.P.FSVS-1}Rtnl1^{CPTI001291}* (RRID:DGGR_115146; *Wakefield and Tear, 2006*), *m12-GAL4* (*Xiong et al., 2010*; RRID:BDSC_2702), *UAS-Rtnl1::GFP* (*Rao et al., 2016*), *{UAS-Xbp1.EGFP.LG}4* (RRID:BDSC_39719; *Ryoo et al., 2007*) and *UAS-tdTomato::Sec61β* (RRID:BDSC_64746; *Summerville et al., 2016*). A second-chromosome insertion of *P{UAS-CG9186::GFP}* (*Thiel et al., 2013*) was mobilized onto the third chromosome using the P transposase source *P{Δ2–3}99B* (*Robertson et al., 1988*; RRID:BDSC_3612), and expressed using *m12-GAL4* in either a $ReepA^+$ or an $Rtnl1^-$ $ReepA^-$ $ReepB^-$ mutant second chromosome background.

For rescue of $Rtnl1^1$ we generated transgenic flies carrying P[acman] clone *CH322-124P15* inserted at *attP2* on chromosome 3 (Bloomington stock 25710), referred to as $Rtnl1^{Pacman}$ (*Venken et al., 2009*). For C-terminal EGFP-LAP-tagging of ReepA and ReepB we used recombineering with the P[acman] system with minor modifications (*Venken et al., 2009*), utilizing the CH322-97D15 (*ReepA*) and CH322-16N11 (*ReepB*) BAC clones (BPRC; http://bacpac.chori.org). An EGFP-LAP-tagging cassette was amplified from R6Kamp-LAP(GFP) (*Poser et al., 2008*) using primers (*Supplementary file 5*) with 50 bp homology to the corresponding *Reep* clone and around 20 bp of homology to the tagging cassette, and used to transform the recombineering *E. coli SW102* strain. Correct clones were verified at every step by PCR and/or restriction digestion. DNA for *Drosophila* transformation was extracted using a PureLink HiPure Maxiprep kit (Invitrogen), and injected into *y w M(eGFP, vasa-integrase,dmRFP)ZH-2A; M(attP)ZH-51D* or *y w M(eGFP,vasa-integrase, dmRFP)ZH-2A; M(attP) ZH-86Fb* (RRID:BDSC_24483 or RRID:BDSC_24749, respectively) at the Department of Genetics embryo injection facility, University of Cambridge, UK. Transformant lines were screened for the presence of the insert by PCR and GFP fluorescence. The primers used are described in *Supplementary file 5*.

BLAST sequence searches were used to define genome coordinates of *P*-element excisions, and Pfam domain coordinates in coding regions, and compare protein divergence rates. They were performed at the National Center for Biotechnology Information (www.ncbi.nlm.nih.gov). REEP

dendrograms were drawn from a ClustalW alignment (*Larkin et al., 2007*) using the neighbor-joining algorithm in MEGA 5.05 (*Tamura et al., 2011*).

## Histology and immunomicroscopy

Third instar larvae were dissected in chilled $Ca^{2+}$-free HL3 solution (*Stewart et al., 1994*), and fixed for 30 min in PBS with 4% formaldehyde. Dissected *Drosophila* preparations were permeabilized in PBS containing 0.3% Triton X-100 (PBT) at room temperature, and blocked in PBT with 4% bovine serum albumin for 30 min at room temperature. Primary antibodies were: Csp (6D6, RRID:AB_10013286, 1:50; *Zinsmaier et al., 1994*), Dlg (4F3, RRID:AB_2314321, 1:100; *Parnas et al., 2001*), (both from the Developmental Studies Hybridoma Bank, Iowa, USA), GFP (Ab6556, RRID:AB_305564, 1:600 Abcam, UK), HRP (P97899, RRID:AB_2314650, 1:300, Sigma), KDEL (Ab50601, RRID:AB_880636, 1:25, Abcam, UK), myc (2272, RRID:AB_331667, 1:25, Cell Signaling, USA). Fixed preparations were mounted in Vectashield (Vector Laboratories, USA, RRID:AB_2336789), and images were collected using EZ-C1 acquisition software (Nikon) on a Nikon Eclipse C1si confocal microscope (Nikon Instruments, UK). Images were captured using 10x/0.30NA, or a 60x/1.4NA oil objective.

## Analysis of ER structure

Confocal images were analyzed blind to genotype using ImageJ (*Schneider et al., 2012*). Images of entire epidermal cells were obtained as z-projections of three consecutive sections. Using the line tool of ImageJ a 12 µm line was drawn from the nuclear envelope towards the periphery of each cell analyzed. Pixel intensity along the line was recorded in an Excel file. Local variance of intensity was calculated by dividing the rolling variance of the intensity (in 10-pixel windows), by the rolling mean intensity, all along the line.

Proximal (anterior) axons were imaged from segment A2, middle images were from the end of segment A4 and A5, distal (posterior) axons were imaged from segment A6 of third instar larvae. Mean gray intensity for single-axon images was measured by drawing a 45 µm line, either along both M12-GAL4-expressing axons (where they could not be separated), or along the most strongly labeled axon (where they appeared as separate axons), and quantifying gray intensity (0–255) by ImageJ; occasional images with saturated pixels were excluded from analysis after blinding. Coefficient of variation was calculated by dividing the standard deviation of staining intensity by the mean; occasional images with faint staining throughout the axon were excluded from analysis after blinding. Gaps were defined as regions where staining intensity was less than 20 (out of 255), after background subtraction.

## Live imaging and FRAP

FRAP experiments were performed on a Nikon Eclipse C1si confocal microscope (Nikon Instruments, UK), using a 20 mW Argon laser and a $40\times$ 0.8 N.A. water dipping objective. Third instar larvae were dissected and incubated in chilled $Ca^{2+}$-free HL3 solution (*Stewart et al., 1994*). Time-lapse images were acquired on a single focal plane every 0.5 or 1 s for 40 loops. For FRAP, a defined region of interest ($12 \times 4$ µm) was photobleached at full laser power (488 nm) for two iterations at a scan speed of 0.5 frame/s. Postbleaching images were acquired on a single focal plane at 15% laser power once every 5 s for 200 s. Experiments were completed within 20 min after dissection. Kymographs were generated for a hand-drawn line selection along the axon using the Multiple Kymograph plugin in Fiji (*Schindelin et al., 2012*). Average fluorescence intensity of each axon in each frame was measured by creating a line selection along the axon. After subtracting background (average intensity in a nearby non-GFP-expressing region within the segmental nerve), the intensity of the bleached region was normalized to the average intensity cross the axon length in the same frame. Then the data were further normalized by taking the prebleaching intensity as 100% and bleach intensity as 0. Normalized postbleaching data were plotted and fitted to a single exponential function to calculate rate constant ($k$) and half time ($t_{1/2}$): $I(t)=A(1-e^{-kt})$, in which $I(t)$ represents the fluorescence intensity at time point $t$, $A$ the highest postbleaching intensity.

## Electron microscopy

For epidermal cell EM, larvae were prepared and fixed as described by *O'Sullivan et al. (2012)*. Transverse sections were cut on a Leica Ultracut UCT ultra-microtome at 70 nm, using a diamond knife, and contrasted with uranyl acetate and lead citrate (for epidermal cells). Sections were viewed using a Tecnai G2 electron microscope operated at 120 kV, and an AMT XR60B camera running the Deben software in the Cambridge Advanced Imaging Centre, School of Biology, University of Cambridge.

For EM of peripheral nerves, we used a ROTO protocol (*Tapia et al., 2012*; *Terasaki et al., 2013*) to highlight membranes. Third instar larvae were dissected in HL3 solution and fixed in 0.05 M sodium cacodylate (pH 7.4) containing 4% formaldehyde, 2% vacuum distilled glutaraldehyde, and 0.2% $CaCl_2$), at 4°C for 6 hr. Larvae were dissected as for confocal analysis, but leaving overlying organs such as gut and fat body attached, to reduce loss of peripheral nerves during processing. Preparations were then washed 3 times for 10 min each at 4°C using cold cacodylate buffer with 2 mM $CaCl_2$. A solution of 3% potassium ferricyanide in 0.3 M cacodylate buffer with 4 mM $CaCl_2$ was mixed with an equal volume of 4% aqueous osmium tetroxide; larval preparations were incubated in this solution at 4°C for 1–12 hr, then rinsed with deionized water at room temperature 5 times for 3 min each. Thiocarbohydrazide solution was prepared by adding 0.1 g thiocarbohydrazide to 10 ml deionized water, kept in a 60°C oven in a secondary embedding pot for 1 hr, swirled every 10 min to facilitate dissolution, and filtered through two 9 cm filter papers just before use. Larval preparations were incubated in thiocarbohydrazide solution for 20–30 min at room temperature and covered with foil to protect from light. Then they were rinsed with deionized water at room temperature 5 times for 3 min each, incubated in 2% osmium tetroxide for 30–60 min at room temperature, and rinsed with deionized water at room temperature 5 times for 3 min each. Preparations were incubated in 1% uranyl acetate (maleate-buffered to pH 5.5) at 4°C overnight and rinsed with deionized water at room temperature 5 times for 3 min each. Then they were incubated in lead aspartate solution (0.66 g lead nitrate dissolved in 100 ml 0.03 M aspartic acid, pH adjusted to 5.5 with 1 M KOH) at 60°C for 30 min and rinsed with deionized water at room temperature 5 times for 3 min. Then they were dehydrated twice with 50%, 70%, 90% and 100% ethanol, twice with dried ethanol, twice with dried acetone and twice with dry acetonitrile. Preparations were incubated in 50/50 acetonitrile/Quetol 651 overnight at room temperature, three times for 24 hr each in Quetol epoxy resin 651 (Agar Scientific, Stansted, UK) and three times for 24 hr each in Quetolepoxy resin 651 with BDMA (dimethylbenzylamine). They were then incubated at 60°C for at least 48 hr.

Serial 60-nm-thick transverse sections were cut in the larval abdominal region, visualized using scanning EM, and images were aligned for analysis of serial sections and reconstruction, as described by *Terasaki et al. (2013)*.

## Axonal EM analyses

To quantify axonal ER tubule diameter, non-axonal staining was removed manually, and ER tubule profiles were identified based on the local threshold in a single cross-section, and the presence of signals at the same position in adjacent sections. The minimum Feret diameter of each tubule was measured using ImageJ Fiji (https://fiji.sc) via the Analyze Particles command. ER numbers per axon were counted manually for all axons detected in the nerve. 3D reconstruction was carried out using the Fiji TrakEM2 plug-in. To quantify gaps in the tubule network, each axon was analyzed throughout the entire stack of sections. To allow for occasional lightly stained or blurred sections, only complete loss of ER tubules from three or more consecutive sections in an axon was defined as a gap. Continuous ER tubules were identified as the presence of signals at the same position for three or more consecutive sections. Given the varying brightness and contrast of EM sections, faint staining that coincided with a tubule signal in adjacent sections was also considered as an ER tubule. Color shading and sketches drawn in *Figure 8* were processed in Adobe Photoshop CS6. For quantification of glial ER sheet length, individual ER sheet profiles were measured using Fiji via the line tool and Measure command. Wrapped axons were defined as one-axon or two-axon fascicles which were completely wrapped by glial cells and isolated from other neighboring axons.

## Statistical analysis

Statistical analyses were performed in GraphPad Prism 6 or IBM SPSS 22. Data were analyzed by two-tailed Student's t-tests or ANOVA followed by post-hoc tests (for comparison of datapoints that were means of raw measurements and hence expected to be normally distributed), or by Mann-Whitney U tests (for data that were not normally distributed). Multiple comparisons after ANOVA were performed by a Tukey HSD test when equal variances were found, or otherwise by Dunnett's T3 test. Bar graphs and scatter plots show mean ±SEM; box plots show median with interquartile range, and the 5% and 95% percentiles as whiskers. Sample sizes are reported in figures. No outliers were excluded from analysis after quantification; data were only excluded from analysis if they were unanalyzable, i.e. when confocal images were too faint or saturated, and EM images lacked clearly identifiable plasma or ER membranes. Most data were analyzed and presented as individual axon or larval datapoints, unpaired between genotypes. For two comparisons where ANOVA showed significant ($p<0.05$) experiment-to-experiment differences within a genotype (*Figures 2B* and *3A*), data were averaged within each independent experiment to produce experiment-wise datapoints, which were compared using two-tailed paired t-tests. $P$ levels are indicated as ns $p>0.05$, *$p<0.05$, **$p<0.01$, ***$p<0.001$, or ****$p<0.0001$, except where indicated otherwise, when criteria could be defined more stringently.

## Acknowledgements

We thank Matthias Beller, Tony Hyman, Ed Levitan, James McNew, Don Ryoo, Zhaohui Wang, the Developmental Studies Hybridoma Bank and the Bloomington, Vienna, Kyoto, and the Japan National Institute of Genetics *Drosophila* Stock Centers, for antibodies, bacterial constructs and stocks. We thank J Skepper and J Powell of the Cambridge Advanced Imaging Centre for help with EM, and S Chan of the Cambridge University Genetics Department *Drosophila* facility for embryo injections. We thank E Avezov, S Chung, P-F Lenne and E Reid for helpful discussions, and Tom Wahlig for his untiring encouragement of our spastic paraplegia work. This work was supported by grants from the Wellcome Trust (08136) and BBSRC (BB/L021706/1) to CJO'K. BY was supported by a Yousef Jameel Cambridge Trust scholarship, LZ by a Marie-Sklodowska-Curie fellowship (660516), NCO'S SZ, and OB by Marie Curie Individual Fellowships (236777, 220851 and 220874, respectively), ALP by a Motor Neuron Disease Association Studentship (861-792), AS by a Pakistan Higher Education Council scholarship, and ZHK by an A*STAR scholarship (BM/RES/07/005).

## Additional information

### Funding

| Funder | Grant reference number | Author |
|---|---|---|
| Yousef Jameel Foundation | | Belgin Yalçın |
| Cambridge Commonwealth, European and International Trust | | Belgin Yalçın<br>Anood Sohail |
| Biotechnology and Biological Sciences Research Council | BB/L021706/1 | Belgin Yalçın<br>Lu Zhao<br>Cahir J O'Kane |
| European Commission | MCSA fellowship 660516 | Lu Zhao |
| Wellcome | 08136 | Martin Stofanko<br>Cahir J O'Kane |
| European Commission | MCSA fellowship 236777 | Niamh C O'Sullivan |
| Agency for Science, Technology and Research | Singapore A*STAR Scholarship BM/RES/07/005 | Zi Han Kang |
| European Commission | MCSA fellowship 220851 | Sophie Zaessinger |
| European Commission | MCSA fellowship 220874 | Olivier Blard |
| Motor Neurone Disease Asso- | Studentship 861-792 | Alex L Patto |

ciation

| Higher Education Commission, Pakistan | Pakistan HEC Scholarship | Anood Sohail |
| UK Hereditary Spastic Paraplegia Support Group | | Cahir J O'Kane |

The funders had no role in study design, data collection and interpretation, or the decision to submit the work for publication.

## Author contributions

BY, Conceptualization, Resources, Formal analysis, Investigation, Visualization, Writing—original draft, Writing—review and editing, Helped write the paper; performed and analyzed most confocal microscopy, epidermal EM, prepared larvae for axonal EM; LZ, Conceptualization, Formal analysis, Funding acquisition, Investigation, Visualization, Writing—original draft, Writing—review and editing, Helped write the paper; performed live imaging and FRAP, ReepB knockdown, analyzed axonal EM; MS, Resources, Formal analysis, Supervision, Investigation, Writing—original draft, Writing—review and editing, Helped write the paper; Generated and molecularly characterized Reep GFP fusions, supervised generation of Reep mutant lines; NCO'S, Formal analysis, Funding acquisition, Investigation, Visualization, Writing—original draft, Writing—review and editing, Helped write the paper; performed and analysed epidermal EM and ER stress; ZHK, AR, MRT, Resources, Investigation, Writing—review and editing, Generated and molecularly characterised Reep mutants; SZ, OB, ALP, Investigation, Visualization, Writing—review and editing, Reep confocal localization; AS, Resources, Visualization, Methodology, Writing—review and editing, Constructed and verified CG9186 stocks; VB, Formal analysis, Investigation, Visualization, Methodology, Writing—review and editing, Performed axonal EM; MT, Conceptualization, Formal analysis, Supervision, Funding acquisition, Visualization, Methodology, Writing—original draft, Project administration, Writing—review and editing, Supervised axonal EM, advised on EM methodology, analyses and interpretation; CJO'K, Conceptualization, Resources, Formal analysis, Supervision, Funding acquisition, Writing—original draft, Project administration, Writing—review and editing

## Author ORCIDs

Lu Zhao, http://orcid.org/0000-0001-7528-4034
Cahir J O'Kane, http://orcid.org/0000-0002-3488-2078

# Additional files

### Supplementary files

• Supplementary file 1. Relative rates of evolutionary divergence of different *Drosophila* Reep proteins, shown by BLASTP searches. BLASTP searches of Refseq proteins, using different *Drosophila melanogaster* Reep protein sequences as queries, show that CG42678 (ReepA) and CG8331 (ReepB) are diverging more slowly than CG4960, CG5539, CG11697, and CG30177, as measured by the change in E value across phylogenetic distance. Each sheet shows the hundred most similar sequences found in each BLASTP search. Hits in *D. melanogaster*, including the most closely related paralogs, are highlighted in yellow.

• Supplementary file 2. Interactive 3D reconstruction of axonal ER from a 4.5 μm segment of a wild-type axon shown in *Figure 7F* (left). Reconstruction is generated from 75 serial 60 nm sections, and shows ER (green), a mitochondrion (magenta), and plasma membrane (gray).

• Supplementary file 3. Interactive three-dimensional reconstruction of axonal ER from a *Rtnl1⁻ ReepA⁻ ReepB⁻* mutant axon shown in *Figure 7F* (right). Reconstruction is generated from 75 serial 60 nm sections, and shows ER (green), a mitochondrion (magenta), and plasma membrane (gray).

• Supplementary file 4. Interactive 3D reconstruction of a 4.5 μm segment of a *Rtnl1⁻ ReepA⁻ ReepB⁻* mutant axon with disrupted continuity of ER, shown in *Figure 7K*. The reconstruction was generated from 75 serial sections of 60 nm each. ER is in green and the plasma membrane in gray.

• Supplementary file 5. Primers used for genotyping, *ReepA* and *ReepB* mutant screening and P[acman] C-terminal tagging. For P[acman] tagging primers, upper case represents sequence hybridising to the gene of interest while lower case represents tagging sequence.

## Major datasets

The following dataset was generated:

| Author(s) | Year | Dataset title | Dataset URL | Database, license, and accessibility information |
|---|---|---|---|---|
| Yalçın B, Zhao L, Stofanko M, O'Sullivan NC, Kang ZH, Roost A, Thomas MR, Zaessinger S, Blard O, Patto AL, Sohail A, Baena V, Terasaki M, O'Kane CJ | 2017 | Research data supporting "Modeling of axonal endoplasmic reticulum network by spastic paraplegia proteins" | http://dx.doi.org/10.17863/CAM.12206 | Publicly available at Apollo - University of Cambridge Repository (https://www.repository.cam.ac.uk/) under a CC-BY 4.0 licence |

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
