## [Decision Letter]

Thank you for submitting your article "Modeling of axonal endoplasmic reticulum network by spastic paraplegia proteins" for consideration by *eLife*. Your article has been reviewed by two peer reviewers, and the evaluation has been overseen by Hugo Bellen as the Reviewing Editor and Anna Akhmanova as the Senior Editor. The following individual involved in review of your submission has agreed to reveal his identity: Vivek Venkatachalam.

The reviewers have discussed the reviews with one another and the Reviewing Editor has drafted this decision to help you prepare a revised submission.

Summary

This paper describes a role, in *Drosophila*, of two REEP proteins that are homologues of human spastic paraplegia disease genes. Specifically, the effects of these REEP homologues on endoplasmic reticulum geometry are categorized using confocal light microscopy and serial electron microscopy. These REEP proteins are shown to reduce ER membrane curvature in axons of *Drosophila* larvae.

The cell biology question of how reticulon and REEP proteins contribute to ER structure and function is important. This paper uses a series of fly lines to gain insight into the roles of axonal ER and the cellular mechanisms underlying axon degeneration disorders. The molecular genetics tools used in this study are impressive. The paper has four new GFP knock in lines and single and double and triple KO lines. The roles of these proteins have been broadly defined in single cell models, but taking this in vivo is an important issue. In addition, the new fly lines are novel disease models, and the work is directly relevant to these human diseases. Previous work has demonstrated the role of a reticulon orthologue in shaping *Drosophila* axonal ER. This work builds on that to describe the individual and combined effects of reticulon and REEP on axonal ER geometry.

The primary conclusion is that REEPA (an orthologue of mammalian REEP1-REEP4) and REEPB (orthologous to mammalian REEP5 and REEP6) work with reticulon to enhance curvature in axonal ER, and also promote the formation of continuous ER tracts over the length of an axon. These conclusions are well-supported by both the light microscopy and electron microscopy data presented.

Major Issues:

A major issue is that it is not clear what the REEP/ reticulon proteins *do* on and *to* the axonal ER.

There are no images showing these proteins in the axons. It is also difficult to see any synaptic ReepA, while ReepB signal is very low (and even so, this is distinct to the issue of axonal-ER). Therefore, defects in axonal ER structure could derive from loss of REEPs/ Rtnl1 in the cell body that, in turn, induces secondary dysfunctions across the cell – a different conclusion to that implied by the authors. Data showing that the ER in the neuronal cell body is normal (if the case, and this is maybe unlikely) would help support the concept that these shaping proteins work directly on the axonal ER. Showing their localization in axons also seems crucial. The fact that ReepA is expressed at very low levels compared to ReepB (based on GFP signal), while only the double KO shows phenotypes, is also surprising and something that the authors could comment on.

On a related note, it is not 100% clear that the defects of axonal ER are cell autonomous to loss of REEP/ reticulons from motor neurons. Indeed, since data in Figure 7 shows defects in surrounding glia. The authors should confirm at the level of assessing motor neuron specific RNAi that the motor neuron phenotypes are cell autonomous (given that MARCM type clonal analysis with combined alleles is difficult).

Another question is whether the EM of Figure 6 definitely assesses "ER". Is it absolutely clear that these structures aren't related to the stalled synaptic vesicles -are these also seen in the TEM?

The authors' conclusion that the ER shaping proteins are required for ER continuity is also not fully supported. This conclusion primarily comes immunofluorescence signal of one smooth ER marker – that could be disturbed while the ER is structurally less affected. The ultrastructural analysis is inconclusive since Figure 6 shows that the% of axons where the ER has gaps is similar between controls and the triple KO genotype. Furthermore, classically this is assessed by live cell photobleaching experiments. Thus, to infer the conclusion that the ER is normally continuous, and this is lost in the disease-model animals, live cell imaging experiments should be attempted and are feasible with larval motor neuron preparation.

How the axonal ER is functionally important / the roles of axonal REEPs/ Rtnl1 also remains less investigated than expected. This is related to one of the points above; it is difficult to dissect without first proving that REEPs / Rtnl1 loss has (at least somewhat) specific effects on the axonal ER. The finding that synaptic vesicle trafficking is stalled is difficult to understand in this respect. Is this something where the axonal ER is particularly important, or could it be a symptom of broad cellular dysfunction?

[Editors' note: further revisions were requested prior to acceptance, as described below.]

Thank you for resubmitting your work entitled "Modeling of axonal endoplasmic reticulum network by spastic paraplegia proteins" for further consideration at *eLife*. Your revised article has been favorably evaluated by Anna Akhmanova (Senior editor), a Reviewing editor, and two reviewers.

The manuscript has been improved but there are some remaining issues that need to be addressed before acceptance, as outlined below:

Can you please address with textual changes or by overexpression studies the comments of Reviewer 3.

*Reviewer #2:*

The modifications made by the authors satisfactorily address all of my previous concerns. I recommend publication.

*Reviewer #3:*

The authors have added significant new data with the photobleaching experiment. I now only have minor comments (stemming from the original critique) that can probably be addressed by modifying the text.

Firstly is the issue of whether Reep/ Rtnl1 proteins are definitely localized in *axonal* ER. I see that Figure 1 shows presynaptic signal, but I still don't see images of HSP proteins in axonal ER that are equivalent to the images showing axonal ER labeled by Acsl::myc (Figure 3 and Figure 4, etc.). It may be that their native levels are too low to detect, although in this case overexpression of tagged proteins would be nice so we at least know that they *can* localize in axonal ER. I feel the authors should explain this lack of data and/ or discuss if it is a detection problem. As written, the paper gives one conclusion: axonal-ER defects are a direct consequence of losing axonal HSP proteins. However, it is also possible that axonal-ER defects occur indirectly because HSP proteins are lost from the main cell body/ synapse. The failure to show them localizing in the axonal ER amplifies this, although I do also see the argument that the presence of axonal ER defects is important to report independent of whether it is direct or indirect to these proteins shaping this particular ER region.

The authors can perhaps add a sentence or two to highlight the data on the cell-autonomous nature of the loss of function defects. Particularly since they show glial abnormalities in the final figure in the genetic background where most experiments are performed.

Since this manuscript has been under review, a new paper on ER structure in neurons was published by Wu, et al., in PNAS. This could be integrated and referenced in the text as it has important data on how we understand ER organization in a neuron (no conflict or overlap with the data here).

---

## [Author Response]

Major Issues:

*A major issue is that it is not clear what the REEP/ reticulon proteins do on and to the axonal ER.*

*There are no images showing these proteins in the axons. It is also difficult to see any synaptic ReepA, while ReepB signal is very low (and even so, this is distinct to the issue of axonal-ER). Therefore, defects in axonal ER structure could derive from loss of REEPs/ Rtnl1 in the cell body that, in turn, induces secondary dysfunctions across the cell – a different conclusion to that implied by the authors. Data showing that the ER in the neuronal cell body is normal (if the case, and this is maybe unlikely) would help support the concept that these shaping proteins work directly on the axonal ER. Showing their localization in axons also seems crucial. The fact that ReepA is expressed at very low levels compared to ReepB (based on GFP signal), while only the double KO shows phenotypes, is also surprising and something that the authors could comment on.*

We completely understand the concern that the reviewers would like to see the relevant proteins at their proposed site of action. However, this is already the case – the statement “while only the double KO shows phenotypes” applies only to REEP phenotypes in epidermal cells (Figure 2), where both genes are expressed (Figure 1); it does not apply to axons, where we see only ReepB::GFP.

In axons, where we detect no ReepA::GFP, loss of ReepA has at most only a marginal effect on ER organisation:

– We detect no phenotype of ReepA mutations on partial loss of ER staining from distal axons (Figure 3);

– We also detect no additional effect of ReepA loss on posterior axons in *ReepA ReepB* mutants, relative to *ReepB* mutants (Figure 3), although there may be an effect on coefficient of variation;

– We have added new data on axonal ER in Rtnl1 ReepB mutants, and see phenotypes that are not significantly different from Rtnl1 ReepA ReepB triple mutants (Figure 4).

In contrast to ReepA, we do detect ReepB::GFP in axons (Figure 1), and much stronger ReepB mutant phenotypes, whether alone or in combination with Rtnl1 loss.

Therefore, we see no disagreement between axonal mutant phenotypes and our lack of detection of ReepA::GFP in axons.

*On a related note, it is not 100% clear that the defects of axonal ER are cell autonomous to loss of REEP/ reticulons from motor neurons. Indeed, since data in Figure 7 shows defects in surrounding glia. The authors should confirm at the level of assessing motor neuron specific RNAi that the motor neuron phenotypes are cell autonomous (given that MARCM type clonal analysis with combined alleles is difficult).*

For reasons discussed in major point 1 above, this applies to Rtnl1 and ReepB.

Rtnl1. We have already shown this in the manuscript:

– In distal axons, similar lowering of ER staining, and similar increase in coefficient of variation of staining intensity, in both M12-GAL4 neuronal knockdown and Rtnl1 mutant (Figure 3);

– No significant difference between Rtnl1 mutant, and M12-GAL4 Rtnl1 knockdown, in the ER fragmentation phenotype of triple Rtnl1 ReepA ReepB loss-of-function motor neurons (Figure 4);

– Note that we previously showed an effect on distal axonal ER, of motorneuron-specific Rtnl1 knockdown, using OK6-GAL4, in O’Sullivan et al., 2012.

ReepB. We have now added data showing that ReepB knockdown in the m12-GAL4 motorneurons leads to partial loss of distal axonal ER, similar to that seen in ReepB mutants (Figure 3).

*Another question is whether the EM of Figure 6 definitely assesses "ER". Is it absolutely clear that these structures aren't related to the stalled synaptic vesicles – are these also seen in the TEM?*

The structures assessed in Figure 6 are tubular – not vesicular. We can follow individual tubules through multiple sections, e.g. see the serial EM sections in Video 5 and Video 7, the resulting reconstructions in Figure 7 (formerly 6F) and [Supplementary-material SD2-data] and [Supplementary-material SD3-data], and in the series of mutant sections in Figure 7 (formerly 6J).

“Are SVs also seen in TEM?” We have drawn attention to some examples in Figure 7 (previously 6I). However, we do not see large accumulations of SVs in EM. This is not necessarily inconsistent with the confocal micrographs in Figure 6 (formerly Figure 5), which show what may only be a single large swollen axon in a ~40µm length of peripheral nerve – our EM reconstructions used only a tenth of this length. We have commented on this briefly in subsection “Rtnl1- ReepA- ReepB-mutant nerves show ER abnormalities in axons and glia”.

*The authors' conclusion that the ER shaping proteins are required for ER continuity is also not fully supported. This conclusion primarily comes immunofluorescence signal of one smooth ER marker – that could be disturbed while the ER is structurally less affected. The ultrastructural analysis is inconclusive since Figure 6 shows that the% of axons where the ER has gaps is similar between controls and the triple KO genotype. Furthermore, classically this is assessed by live cell photobleaching experiments. Thus, to infer the conclusion that the ER is normally continuous, and this is lost in the disease-model animals, live cell imaging experiments should be attempted and are feasible with larval motor neuron preparation.*

First, we have also used a new marker, a smooth ER lipase, CG9186 tagged with GFP, to show continuous labeling in wild-type axons, but occasional gaps in *Rtnl1 ReepA ReepB* triple mutants, consistent with the observations using Acsl::myc (new Figure 5).

Second, we have generated movies of live larval axons carrying this marker, and carried out photobleaching. We find that ER organisation in axons is moderately stable, albeit with some dynamic features. We also find that recovery from photobleaching is rapid, and normally occurs from both ends of the bleached region. However, when we find a gap in CG9186::GFP labeling in triple mutants and photobleach next to it, there is no recovery from the end with the gap (Figure 5, Figure 5—figure supplement 1, and Video 1–Video 4). Therefore we conclude that the ER network is continuous over the lengths that we are analyzing, and that gaps in staining of ER markers also represent physical gaps in the network.

Our live imaging has also given us insights into the stability and dynamics of the ER network in axons; these data are also presented in Figure 5, in Figure 5—figure supplement 1, and in Video 1-4.

*How the axonal ER is functionally important / the roles of axonal REEPs/ Rtnl1 also remains less investigated than expected. This is related to one of the points above; it is difficult to dissect without first proving that REEPs / Rtnl1 loss has (at least somewhat) specific effects on the axonal ER. The finding that synaptic vesicle trafficking is stalled is difficult to understand in this respect. Is this something where the axonal ER is particularly important, or could it be a symptom of broad cellular dysfunction?*

There seem to be a few issues rolled into one comment here, and we will do our best to address them individually.

– “…*Proving that loss of REEPs/Rtnl1 has (somewhat) specific effects on axonal ER*”. In our answer to point A above, we emphasise that these proteins are found on the ER, where their loss has an effect; they do not show the same distrobution as synaptic vesicles (e.g. O’Sullivan et al., 2012), which mostly fill out the presynaptic bouton but are barely detectable in axons. Furthermore, the mutant phenotypes are exactly as predicted from the extensive biochemistry already published on these proteins – loss of ER membrane curvature. Both of the localization and the biochemistry strongly support the model that Rtnl1 and REEPs have specific effects on ER.

– “*Is axonal ER particularly important for SV trafficking, or could there be broad cellular dysfunction?* The triple mutants are viable as adults, and with some care we can even maintain them as a homozygous stock, although their adult lifespan is halved to around 2 weeks from controls. The accumulation of SV markers in axons is also relatively mild compared to some other reported mutants, e.g. Dhc and Appl (Gunawardena et al., Neuron 32:389), Acsl (Liu et al. 2011, J Neurosci 31:2052), Khc/KIF5a (Füger et al. 2012, PloS Gen 8: e1003066). Taken together, any broader cellular dysfunction must be relatively mild. We have already discussed some of the cellular consequences of abnormal ER organisation in the Discussion section, and have added the axon transport issue to this discussion.

*– How axonal ER is functionally important… remains less investigated than expected*”. The functional consequences for axons are a very broad question, which we believe is for another day, given that we already have a complete manuscript with new and well supported findings, the large amount of data we already present, the extent to which our findings and reagents will support future research on this question, and the fact that exploring the functional consequences is a large undertaking in its own right.

[Editors' note: further revisions were requested prior to acceptance, as described below.]

[…] Can you please address with textual changes or by overexpression studies the comments of Reviewer 3.

*[…] Reviewer #3:*

*The authors have added significant new data with the photobleaching experiment. I now only have minor comments (stemming from the original critique) that can probably be addressed by modifying the text.*

*Firstly is the issue of whether Reep/ Rtnl1 proteins are definitely localized in axonal ER. I see that Figure 1 shows presynaptic signal, but I still don't see images of HSP proteins in axonal ER that are equivalent to the images showing axonal ER labeled by Acsl::myc (Figure 3 and Figure 4, etc.). It may be that their native levels are too low to detect, although in this case overexpression of tagged proteins would be nice so we at least know that they can localize in axonal ER. I feel the authors should explain this lack of data and/ or discuss if it is a detection problem. As written, the paper gives one conclusion: axonal-ER defects are a direct consequence of losing axonal HSP proteins. However, it is also possible that axonal-ER defects occur indirectly because HSP proteins are lost from the main cell body/ synapse. The failure to show them localizing in the axonal ER amplifies this, although I do also see the argument that the presence of axonal ER defects is important to report independent of whether it is direct or indirect to these proteins shaping this particular ER region.*

– We have added UAS-Rtnl1::GFP to Figure 1, showing localisation similar to that of Acsl::myc driven by the same motor neuron GAL4 line.

– We cannot detect ReepA::GFP in axons, but we find at most a subtle contribution of ReepA function to axonal ER in Figure 3 and Figure 4, and therefore see no inconsistency here.

– We do not have a UAS-ReepB::GFP construct, but have provided further images showing localization of ReepB::GFP to the axonal region in a peripheral nerve (single confocal section in Figure 1) and in axons as they emerge from nerve branches to reach the NMJ (Figure 1).

*The authors can perhaps add a sentence or two to highlight the data on the cell-autonomous nature of the loss of function defects. Particularly since they show glial abnormalities in the final figure in the genetic background where most experiments are performed.*

Done (third paragraph of Discussion).

*Since this manuscript has been under review, a new paper on ER structure in neurons was published by Wu, et al., in PNAS. This could be integrated and referenced in the text as it has important data on how we understand ER organization in a neuron (no conflict or overlap with the data here).*

Added to Discussion, including the possibility that we might be missing occasional “thin ER” in our preparations.